# Architectural Systemic Approach: The Serpentine Gallery 2005, a Reciprocal Frame Case Study

**Beatriz del Río-Calleja \*, Joaquín Grau Enguix † and Alfonso García-Santos**

Department of Construction and Architectural Technology, Escuela Técnica Superior de Arquitectura de Madrid, Universidad Politecnica de Madrid, 28040 Madrid, Spain; alfonso.garciasantos@upm.es
* Correspondence: beatriz.delrio@hotmail.com
† Deceased.

**Abstract:** The application of the systemic approach in architecture aims to promote an integral, holistic view of the architectural design process. The literature reviewed calls for models with systemic behavior, and for these models to be applied in concrete cases. This paper proposes an original approach, using the foundation matrix and the constructive logic matrix. Both matrices are part of a developing model that is being tested on a case study. The work presented here had two objectives: to check this part of the model and gain more knowledge about the model itself. The selected case study, the 2005 Serpentine Gallery Pavilion, is a contemporary ephemeral construction of significant architectural interest. It is a reciprocal frame structure, linked to the construction history. The methodology used was a systemic analysis. In the first phase of the analysis, the reciprocal structures documented historically in the West were reviewed. The other two phases corresponded to the application of the two model matrices. Conceptual diagramming was used in all phases of the process. The results show the importance of the study of historical building solutions. The use of matrices facilitates the identification and understanding of the operations carried out in the design process of the case study. Matrices favor the organization of concepts and relationships from through a systemic approach. Understanding generation operations in an integrated way leads to a type of knowledge (relational knowledge) that allows architecture to be thought about in a holistic way. This makes the systemic view of art and technology as a unit possible, attending to the whole complexity of architectural thinking.

**Keywords:** systemic approach; holistic view of architecture; project complexity; construction history; relational knowledge; thinking process; reciprocal frame structures

## 1. Introduction

The systemic approach consists of conceiving a whole in an integral way, whose related parts form an entity of a higher order than the sum of the isolated parts. Systemic analysis is the usual tool for understanding complex systems [1].

The application of the systems approach to architecture and the construction industry has given rise to theses and articles that can be divided into two groups. On the one hand, foresight studies have demonstrated the benefits of addressing architecture and the construction industry by using this systemic approach. Is a useful approach for understanding complex processes inherent to an architectural project, the construction industry, and integrated knowledge management [2–14]. On the other hand, there is systemic research with a practical character in architecture and construction. Models oriented toward solving complex problems have been developed. Most of the practical applications are directed toward the search for production processes and assembly systems oriented toward more flexible contemporary industrialization [15–22].

A review of the scientific literature reveals that in the research on new methodologies to address the complexity of the building process and the architectural design process,

there seems to be common ground. This common basis is multifaceted and encompasses the systems approach, complexity theory, dynamic adaptive systems, and the holistic view. Finding a common theoretical basis between these disciplines may be the first step toward tackling the problem [1].

The scientific community is pursuing this line of research based on evidence that the building process and the architectural design process can be understood as a complex system. The application of these theories in architecture is still emerging. More research is required.

In the scientific literature, some methods have been developed to approach architecture from a systemic viewpoint. However, while most define a methodology, only a few go as far as applying it to concrete cases. They usually suffer from using a model that has no real systemic behavior [19,22,23].

What is essential in a systems approach is to attend to the definition of the parts through the relationships that are established between them. The authors of a recent paper addressed the scientific community's call for theoretical background. General systems and complexity theory, design methods, and design thinking have been considered [24]. Through this review and the diagramming presented, a common theoretical basis to these disciplines is put forward.

To address the gaps identified in the scientific literature, it is necessary to develop research in which the systems approach is applied to a concrete case study. The process for doing so must ensure that the systemic behavior of the approach is maintained, enabling a holistic vision, relational analysis, iteration, evolution, adaptability, and adjustment.

This paper presents the results of a systemic analysis of a case study, the Serpentine Gallery Pavilion 2005. The analysis strategy combines diagramming, analysis and synthesis, and an original approach through what have been defined as systemic analysis matrices. We have defined and used two matrices. One, called the foundation matrix, consists of the sequence of the concepts: idea, form, and image. The second, called the constructive logic matrix, is made up of an interrelation of the concepts: geometry, material, and function. Both matrices are developed in the methodology section and prove useful in the systemic approach taken to building the case study.

### 1.1. Serpentine Gallery

Since 2000, the Serpentine Gallery team has been building a pavilion in Kensington Gardens in London every summer. It is an ephemeral installation, the aim of which is to display the best international contemporary architecture in a genuine way. The so-called "Summer Pavilions" are laboratories for architectural experimentation via small works in which architects freely express their ideas but that may also constitute a research proposal with a formal, material, and structural character [25,26].

For this case study, the 2005 Serpentine Gallery pavilion by Alvaro Siza, Eduardo Souto de Moura, and Cecil Balmond has been chosen. The 2005 Serpentine Gallery is an ideal case study because of its scale and architectural interest. It was a reciprocal timber structure linking a contemporary proposal to the construction history [27]. Cecil Balmond, whose design approach engages inner organizational systems, intervened in its formal and structural definition [28].

Cecil Balmond advocates a holistic view of architecture, integrating art and science. He revises the meaning of geometry, form, and structure through dynamic systems. Until 2011, he was Vice President of Arup and headed the Advanced Geometry Unit (AGU) [29–31].

### 1.2. Reciprocal Frame Structures

The terms "reciprocal networks"/"reciprocal frames"/"reciprocal structures" are now widely used to designate structures created by arranging short linear elements around a pattern so that they mutually support each other and cover a larger span. The pattern forms a stable geometric configuration in which the elements share and transmit the load to the supports [32–34].

This principle of reciprocity in the design and construction of structures has been known since antiquity [32,35]. It is included in the main construction treaties. A chronological review of the use of the principle of reciprocity throughout the history of construction was carried out. The review aimed to see how it has evolved in the West, assess the continuity or discontinuity of the use of this principle as a construction solution, and question the value of the history of construction as applied to contemporary architecture.

We list the documents and projects reviewed: Villard de Honnecourt "*Livre de Portaiture*" (first half of the 13th century), Leonardo da Vinci "*Codex Madrid I*" or "*Tratado de estática y mechanica*" (1490–1508) and "*Codex Atlanticus*" (1478–1519), M. Sebastiano Serlio "*Il primo libro d´Architettura*" (1545), Philibert de l´Orme "*Nouvelles inventions pour bien bastir et á petits frais*" (1561), Johanne Wallis "*Mechanica: sive, De motu, tractatus geometricus*" (1670), Frézier "*La théorie et la pratique de la coupe des pierres et des bois*" (1738), Fourneau "*L'art du trait de charpenterie*" (1770), Krafft "*Plans, coupes et élévations de diverses productions de l'art de la charpente*" (1805), "*Traité sur l'art de la charpente théorique et pratique*" (1819), Rondelet "*Traité théorique et pratique de l'art de bâtir dedicates the fourth tome to carpentry*" (1810), Amand Rose Emy "*Traité de l'art de la charpenterie*" (1837), Thomas Tredgold "*Elementary Principles of Carpentry*" (1890), Giuseppe Antonio Borgnis "*Traité élémentaire de construction appliquée à l'architecture civile*" (1823), Nicola Cavalieri San Bartolo "*Istituzioni di Architettura Statica e Idraulica. Volume I*" (1826), Valadier "*L'architettura pratica dettata nella scuola e cattedra dell'insigne Accademia di S. Luca. Tomo II*" (1831), Mazzocchi "*Trattato su le costruzioni in legno*" (1871), Curioni "*L'arte del fabbricare. Lavori generali di architettura civile stradale ed idraulica*" (1868), Caveglia "*Corso di costruzioni civili e militari. Volume terzo*" (1878). Since the beginning of the 20th century, until the 1950s, reciprocal structures have been mentioned in all Italian building treatises and manuals [35,36].

From the mid-20th century onwards, we find examples in projects and works, among which the following stand out: the *Lamella wood system* by Zollinger (1920s), *Casa Negre* by Josep Maria Jujol (1915–1925), the *cité de circulation* designed by Theo van Doesburg (1924–1929), the *Mill Creek Housing Project* designed by Louis Kahn (1952–1953), the *Berlin Philharmonic* by Hans Scharoun (1960–1963), the *Rice University bamboo canopy* by Shigeru Ban and Cecil Balmond, the *Forest Park Pavilion* by Shigeru Ban and Cecil Balmond (2004–2007), the *Serpentine Gallery 2005* by Siza, Souto de Moura, and Balmond (2005), and the *Rokko Observatory* by Sambuichi and Arup (2010).

Pugnale took a historical and critical look at the principle of reciprocity in construction [35]. They reviewed the historical background of reciprocal structures in Western and Eastern culture. They listed the registered patents and established a classification of the growing number of scientific publications from the past decades with seven groups: morphology and geometry; form-finding and morphogenesis; joints and connections; structural behavior; the kinematic potential to generate adaptive structures/architectures; material and sections' realization; discussing architectures, constructions, and prototypes.

## 2. Methodology and Model

A systemic analysis is a suitable interdisciplinary methodology for the study of complex processes, taking the environment, behavior, and mechanisms of the system itself into account.

To analyze systemically means considering that the parts of a whole and their relationships form a higher-order (integral) entity than that formed by the sum of the isolated parts. Conducting a systemic analysis is complex, because as the elements that make up the whole emerge, so do the relationships between them. This could be described as a simultaneous process of analysis and synthesis. Claeys proposed that the method should allow analysis and synthesis to be complementary in the elaboration of complex thinking [1].

Above all, importance should be given to the interactions between the relationships of the parts as it is these (the relationships) that give identity not only to the parts but also to the whole.

Methodologies that approach architectural design as a process have two basic stages in common: analysis and synthesis. These two activities operate in tandem and iteratively. Analysis and synthesis occur simultaneously and cyclically. There are models that try to graphically represent how these two operations are carried out simultaneously. The most widespread are: *Diverge/Converge vs. Narrow/Expand*, *Oscillation*, and *Diverge and Converge Cycled* (Figure 1).

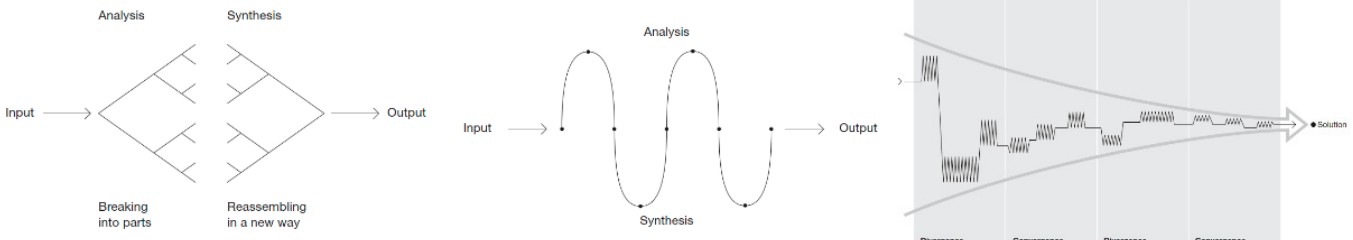

**Figure 1.** Design model schemes based on Christopher Alexander´s and Nigel Cross´s works [37].

### 2.1. Graphical Tools

Conceptual diagramming is used as a graphic tool. While verbal language is sequential, graphic language is simultaneous; all symbols and their relationships are considered at the same time, making it possible to visualize sets. The process is to record ideas and functions, facilitating the visualization of relationships within the whole. It is an essential thinking tool because it helps to understand the structure and behavior of the system and identify the hierarchy of functions [38].

Diagrams or concept maps are a tool for the structured, organized representation of knowledge [39]. Systems take the form of networks or graphs. The human eye is a remarkably powerful analytical tool and looking at images of networks is an excellent way to understand their structure [40]. The nodes represent the concepts and links between nodes are the relationships. Blackwell described a diagram as an analytical statement, i.e., a set of illustrative marks and written notes that governs and transforms the meanings of verbal statements into a graphical context useful for solving complex problems [38]. Graphs show at a glance relationships that are more difficult to see in linear notations.

The conceptual diagramming (mapping) of information supports the understanding of the organization or configuration that will define the principles and criteria for decision-making throughout the process.

The diagramming used in this research allowed for a complementary analysis and synthesis. It facilitated the separation of the whole into parts, and at the same time, enabled us to visualize the relationships between them. Diagrams have been developed of the historical evolution of reciprocal mesh construction solutions, the foundation matrix (idea, form, and image), and the construction logic matrix (geometry, material, and function).

### 2.2. Systemic Analysis Process
#### 2.2.1. Phase 1_Documentary Phase

In this phase, information was gathered on two topics: on reciprocal structures throughout construction history and about the case study itself, and its specific design and construction process.

The case study is of a reciprocal frame structure. Therefore, the historical development of this solution was reviewed, as stated in the introduction, identifying geometric patterns that help us to understand the Serpentine 2005 structural geometry. The historical review allowed us to make a chronological reading of the solutions.

To carry out the systemic analysis of the case study (Serpentine Gallery 2005), the information available on the building was reviewed to provide information on the development process and definition of the project, its construction characteristics, and its execution.

### 2.2.2. Phase 2_ Diagramming and a Chronological Analysis of Reciprocal Frame Structures

Diagramming was used as a graphical tool to elaborate the historical analysis. A chronological diagram of the reciprocal structures was drawn up from the information gathered in phase 1 on the use of the principle of reciprocity throughout the construction history.

The author and date are indicated in the chronological diagram. The solutions were redrawn, showing for each one the proportions, types of union, and geometric pattern of the mesh. The historical review enabled us to understand how they were adapted and evolved throughout the construction history.

### 2.2.3. Phase 3_ Diagramming and Systemic Analysis: (FM) Foundation Matrix—Idea, form, Image

By developing conceptual diagrams, a systemic analysis of the information obtained from the case study in phase 1 was carried out. The aim was to structure and organize the information around a matrix made up of three components: the idea, the form, and the image (foundation matrix). An in-depth philosophical discussion of the concepts of idea/form/image is not the subject of this article. Considering this sequence as the organizer of a plan that establishes and guides the development of an architectural project from its conception to its execution.

The foundation matrix consists of idea, form, and image, key concepts in the context of creation and design. The creative process evidences a path that leads from the intelligible to the sensible reality and on which we sequence the concepts of idea, form, and image.

The idea is always unity, but it appears in multiple ways. It is a mental construction that requires a period of work for its elaboration and belongs to the field of the intelligible. This does not mean that sensitive elements coming from reality are not considered in its elaboration.

From the diagramming and analysis of the idea based on this approach to the documentation collected on the case study, a configuration of the objective was obtained from the idea as an organized unit.

In the elaboration of an idea as a mental construct, it is necessary to identify the elements that make it up. Diagramming makes it easy to identify these elements. The idea results from a study of the previous conditions and brings together the objectives to be achieved in the design, which we summarize in what is sought, what is intended, and its essence. The unitary vision of these elements allows the configuration of the objective as an organized unit. A unitary vision of the idea as a plan-former is obtained, an orientation that governs the decisions and creative process.

The diagramming, systemic analysis, and study of the parts that configure or shape the idea (understanding of the objective) show the relationships between elements. Fields of knowledge also become the object of study to deepen the concepts derived from the development of the idea.

The form is a configuration that supports matter. Following the defined sequence (idea, form, and image), the idea uses the form as a support for the matter, with the intention to achieve a sensible reality.

The form, as a configuration, has an internal structure. It is this internal structure that makes the adaptability attributed to the concept of form possible. The form adapts at each stage of the design process and has the capacity to reach dynamic states of equilibrium. This dynamic equilibrium means that, in the face of certain variables, there is a capacity for adjustment. The capacity is maintained if the preset variables change, resulting in another adjustment. The shape is a dynamic equilibrium that allows for the consideration of parameters or variables through a process of adjustment maintained over time. Form is inherent to the design process as it evolves and transforms throughout. The form, therefore, is inherent to the process followed for its generation, its creation. In the design process, the aim is to achieve dynamic equilibrium. The form can adapt, adjust, and reach equilibrium due to its internal configuration. It is the internal structure of the form that helps us to

realize the potential for adjustment, adaptation, and dynamic equilibrium between its predetermination and the possible configurations it can assume.

In the idea-form-image sequence, the form arrives at the image through identification. Understanding that to identify oneself is to make an effective personal imprint is to accept what has been created by bringing the form into existence. This sequential approach necessary to reach a definition of the image was taken by authors such as Peter Zumthor, who argued that it is the (abstract) idea that leads to the image [41].

The methodology holds that the image is an end product of the design process. It is also necessarily linked to personal identification with the work. Analyzing the image as an end product does not provide information about the design process or the construction process and is therefore not done.

In the proposed foundation matrix, there is a strong linear sequence between the concepts of idea/form/image when they are treated independently. The three concepts interrelate with each other, but in the case of this matrix, they always do so from the overall level. We have defined a higher hierarchical level in which this interrelation between the three concepts is considered. This global level to which we refer is represented graphically in the foundation matrix by the vertex of the tetrahedron in Figure 2 and Figure 11.

### 2.2.4. Phase 4_ Diagramming and Systemic Analysis: (CLM) Construction Logic Matrix—Geometry, Material, Function

Systemic analysis is carried out from a perspective that we call constructive logic, bringing together considerations of material, function, and geometry. This phase considers the relationships revealed in phase 3.

The construction logic matrix considers a system in which all elements are holistically related according to the three perspectives considered (geometrical, material, and functional) and at the different scales (source-material, element, building). There is always reference to the building and the possibility of establishing a system of relationships that allows for the generation of the elements and the whole, in an integrated way; for this to be possible, structures at different scales and orders must be related.

This triple referential aspect presents a certain independence and makes integration possible. Thus, it is possible to realize the geometric approach to the case study, knowing that it will be operationally related to the geometrization of the function and to the geometric understanding of the material and its transformation operations. In this way, one variable can be resolved through the determinations of the others [42].

The matrix of constructive logic is formed by function, geometry, and material. Unlike the foundational matrix, this second matrix does not operate sequentially, and its diagramming is more complex. The proposed approach considers a system in which the components are holistically related. The definition of variables from one of the components has consequences and defines the boundaries of the other two components. The definition and its implications are back and forth paths that operate in a systemic and iterative way.

We consider three perspectives (geometry, material, and function) and three orders (source-material, element, and building) [43].

The order in the understanding of the configuration system rests on the structural organization of the building. The testing of the systemic approach focuses on the roof element, which is considered to be the most complex element.

In terms of the three perspectives:

- Geometry supports deduction. It allows relationships to be established between the parts, and it helps us to realize the unit as a whole to be mastered. This deductive nature of geometry constitutes the support system for the generation of constructive entities;
- Material is the possibility. It is the path to the realization of the idea;
- From a function perspective, systemic analysis seeks to define the behavior of the elements that make up the case study. This logic underpins the study of the functions and the relationships between them.

### 2.3. Systemic Analysis Matrices Workflow

To represent the workflow of systemic analysis matrices graphically, it is necessary to propose a model. Models are constructs designed by an observer to identify and measure complex systemic relationships. Every real system has the potential to be represented in more than one model.

Each of the systemic analysis matrices defined in this paper has three components. The use of triangular structures is proposed to support their development and understanding, then they can easily create three-dimensional structures by repeating the triangular ones. The triangle is a stable geometric figure, and one of the polyhedrals that can be formed by repeating the triangle on all its faces is the tetrahedron. As a three-dimensional figure, the tetrahedron can generate networks or meshes. The choice of the tetrahedron as a support structure is a way of ordering the concepts that make up each matrix. It allows both the components and the relationships between them to be organized, and at the same time, gives certain independence to each of the matrices. In this way, the use of this spatial polyhedral structure allows for the study of the concepts and the matrices both in isolation and combined.

The representation of the matrices through the tetrahedron is influenced by the graphical representation of the mathematical model of the finite automata. The arrangement of the terms of the matrices facilitates the identification of the relationships between them. The relationships between the components of the foundation matrix (FM) are sequential. The idea is related to the form (x) and the form to the image (y). The three components are, in turn, related on an overall level (A, B, C), which is represented by the vertex of the tetrahedron in the model (Figure 2). In the constructive logic matrix (CLM), the relationships between the components are systemic. They are all related to each other (x, y, z), and in turn, as a whole (A, B, C) (Figure 2). The relationships between the two matrices are through the vertices of the tetrahedron; they are related at an overall level (dashed line in Figure 2).

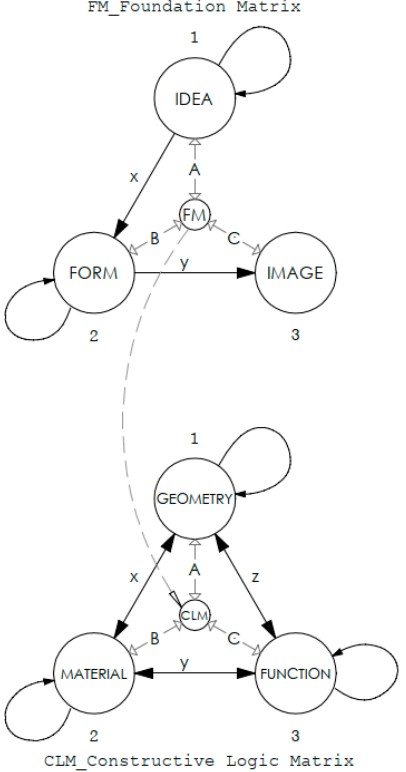

**Figure 2.** Matrices model and workflow.

## 3. Results and Discussion

### 3.1. Constructive Historical Analysis of Reciprocal Frame Structures

The chronological diagram (Figures 3–5) enables a reading of the evolution of reciprocal frame structures as a constructive solution in the West. It shows the adaptation, distortion, and adjustment of the basic module or fan over a long period, ranging from the 13th to the 21st century.

Analysis of the constructive solution of floors and roofs (flat or pitched), using elements of smaller dimensions than the span to be bridged, allows us to recognize the basic module or fan. First documented by Villard de Honnecourt [44], and currently called a fan, it is usually made up of four elements, although the minimum number is three. This analysis makes it possible to recognize and distinguish different proportions between the constituent elements of the fan in the documented solutions (Figures 3–5).

The earliest documents date back to the Middle Ages [44] and early Renaissance [45–47], in which the dimensions and proportions of solutions did not cover large areas (Figure 3a–e).

Graphical analysis of the solutions reveals the strategies and operations carried out on the basic module, among which operations such as rotation, symmetry, displacement, and/or repetition can be distinguished (Figures 3–5). From the basic module and using one or more of the identified operations, larger solutions appear to cover larger spaces (Figures 3 and 4). The solutions adapt and evolve [48] in the resolution of larger-area mesh sizes, such as those defined by Wallis [49], who studied the principle, carried out an analytical calculation, and defined patterns with a base module made up of a greater number of elements. Wallis develops mesh solutions in which the rotation and symmetry operations are already present, where the span covered is four times the length of the larger element (Figure 3g–k). In the 18th century, Frézier [50] became interested in the principle (known as Serlio's floor) and linked it to the constructive solution of Abeille's flat vaults. The establishment of this relationship gave rise to numerous articles that jointly reviewed both solutions [51–53].

In the 18th and 19th centuries, treatises on carpentry included constructive solutions based on the principle of reciprocity. These were mostly oriented toward the construction of floor slabs. Fourneau [54] referred to questions of size and proportion (Figure 4a). The slab thickness is significantly reduced and only joists are used. For a span of 5.85 m, thicknesses between 8 and 11cm are used, thus obtaining a ratio between thickness and span of 1/72 and 1/51. The norm at the time, according to Rondelet, was a thickness of 1/24 for the span. Krafft [55,56] considered the role of the planking to be essential for good structural performance and recommended double-layer planking laid with crossed directions and staggered joints (Figure 4b). Rondelet [57] described how to construct a Serlio floor using different types of joints. He defined the dimensions of the elements and the joints and developed an example using a single-base module or fan, reinforcing the joints with iron nails and strips (Figure 4c). Tredgold [58] maintained that it is a curious solution, and one which does not go unremarked. Yet, he considered that its use was infrequent as long timber was usually available. For Amand Rose Emy [59], the principle of reciprocity was imitated by an amusement found in an old collection of mathematical recreations, where three or four knives we placed so that the ends of the handles rested on fixed points, arranged so that their alternately crossed blades could hold an object in the air (Figure 4d). Emy showed the solutions described by Krafft and Rondelet. Mazzocchi [60] analyzed all the examples of Krafft, Rondelet, and Emy. He encouraged the use of iron elements as reinforcement and underlined the role of the planking. The solutions considered the types of joints between elements, distinguishing between mortise and tenon, dovetail, metal joints (use of pins or nails), or direct support of some elements on others (Figures 3–5).

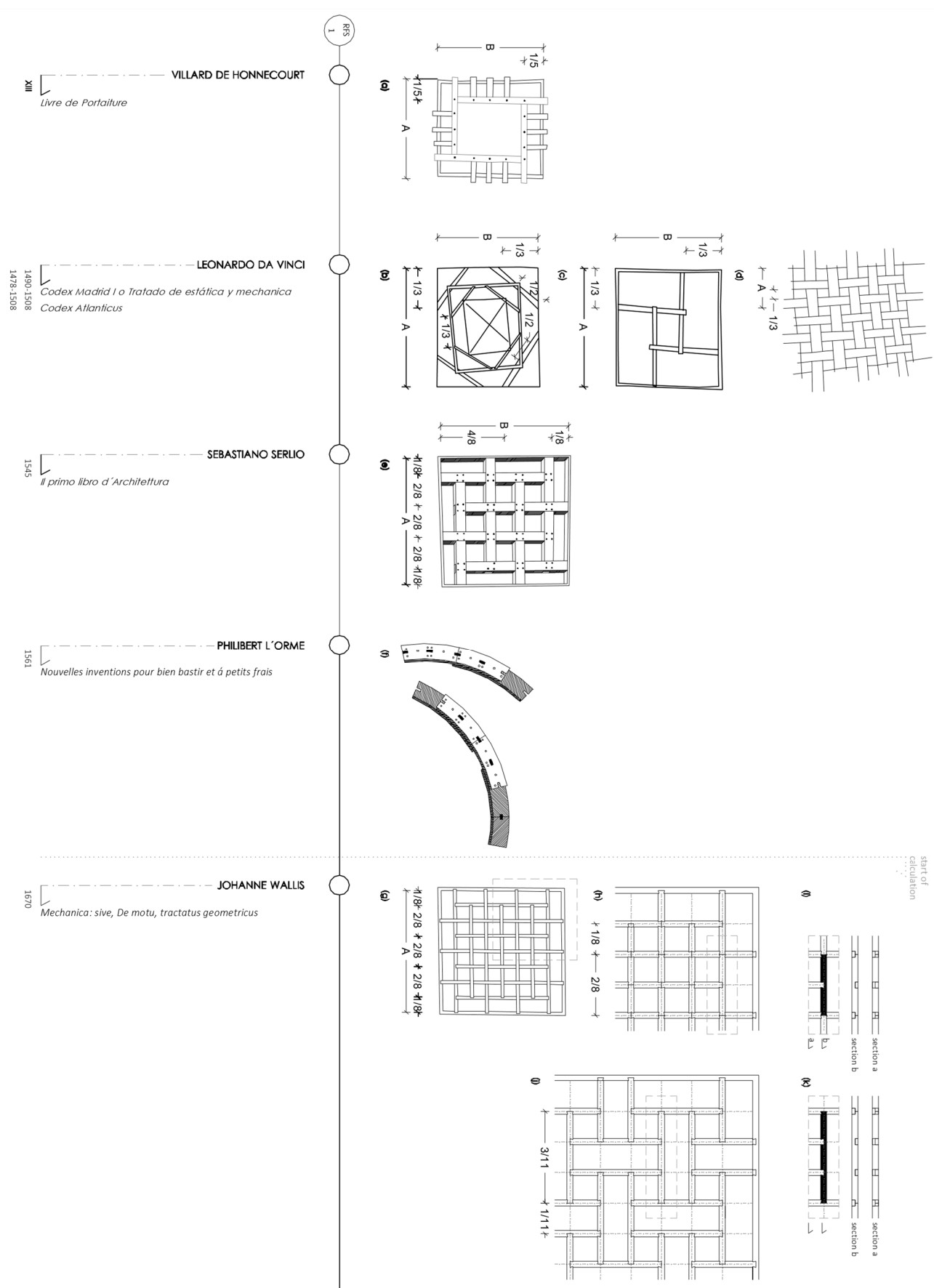

**Figure 3.** Chronological diagram of reciprocal frame structure 1.

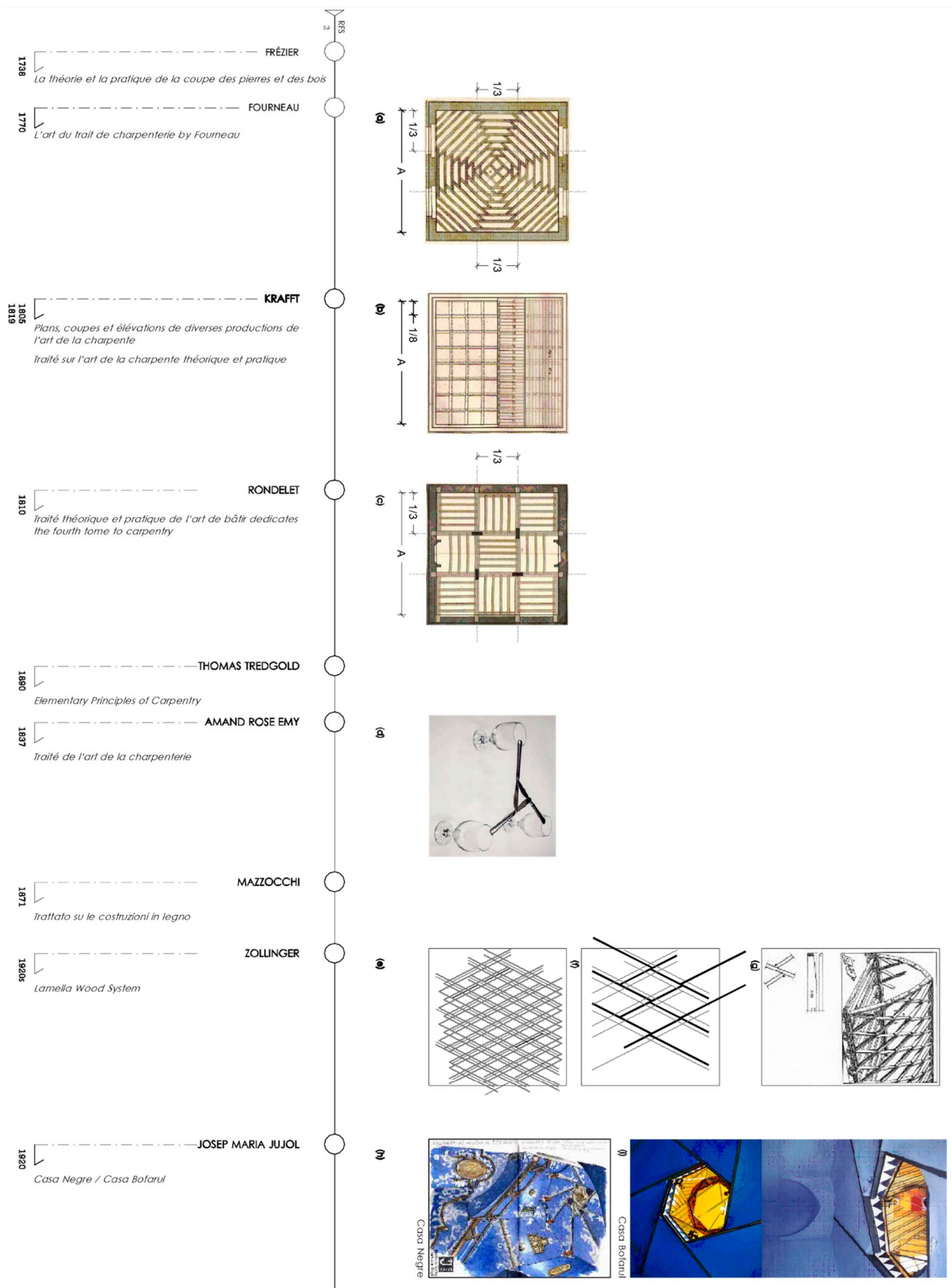

**Figure 4.** Chronological diagram of reciprocal frame structure 2.

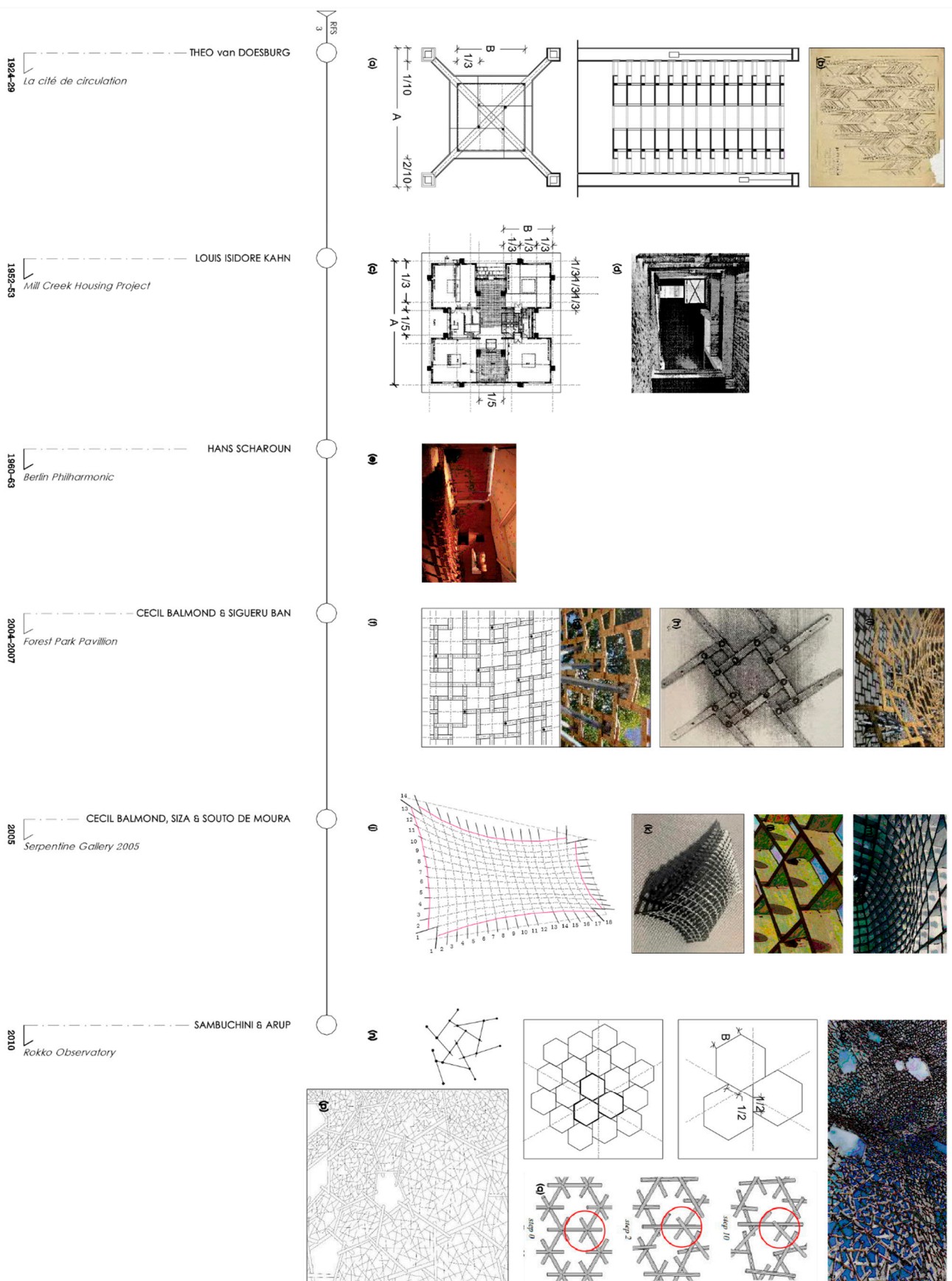

**Figure 5.** Chronological diagram of reciprocal frame structure 3.

In the 20th century, in 1924, Zollinger [33] filed his patent that adapted the orthogonal mesh (the most frequent solution in documented historical cases) by proposing a rhomboidal and inclined mesh for the construction of steeply pitched gable roofs and for shaping a roof profile in the form of a pointed arch. This introduced a novelty to the resolution of the joint [61], displacing the elements that reach the main one, thereby preventing them from coinciding at the same point, which favors the construction of the structural system (Figure 4e–g). This joint minimizes the number of shared meeting points, and the interwoven beams give strength to the structure [62]. The solution has similarities with the construction of arches using small elements, as developed by Philirbert d'Orme [63] in his 16th-century treatise (Figure 3f).

Our chronological analysis included built examples, such as that designed by Jujol [64], in which the elements rested on top of each other, forming a fan of six elements. The geometry of these generated a sloping surface of revolution that served as a roof, bearing similarities with common solutions in the East (Figure 4h,i)).

Reciprocal frame structures are generally made of built-in timber, but they were translated into the language of reinforced concrete in the 20th century. There are examples designed by Theo van Doesbourg [65], Louis Kahn, and Hans Sharoum [66] (Figure 5a–e).

The most recent examples (Figure 5f–q) have, as a common feature, the involvement of Arup in the constructive definition. The patterns are based on those defined by Wallis and are adapted or distorted to resolve curved surfaces. The constructive resolutions are executed with wood, in particular, bamboo at the Forest Park Pavilion [30] (Figure 5f–i) and micro-laminated timber at the Serpentine Gallery [67] (Figure 5j–m). In the case of the Rokko observatory [68,69], the main structure of the mesh that forms a distorted dome is made of a circular section of steel tube, and cedar wood is only used to make the lattice pattern of each hexagon (Figure 5n–q).

A chronological reading of the use of this structural principle shows that there has been a certain continuity, marked by the irruption of reinforced concrete and steel at the beginning of the 20th century. Though there were attempts to transfer reciprocity to reinforced concrete, those were not very significant. From the 21st century, we can find examples of singular architecture in which this type of structure has been used. The structural principle is the same, but its geometry and form are complex; the meshes curve, distort, and adapt to new formal intentions. The use of contemporary materials, often produced via prefabrication systems, makes it possible to cover larger surfaces.

*3.2. Systemic Analysis of the Foundation Matrix*

The diagrams should be read as a whole, noting their tree structure. The dashed lines signify the relationships between the concepts. To assist in reading the diagrams, the areas corresponding to a heading have been marked with Roman numerals. Areas corresponding to drawings are marked with capital letters. Images are indicated by lowercase letters in brackets, and bibliographical references with numbers in square brackets.

The idea may be shaped into an organizing plan through comprehension of the target (Figure 6I) and by questioning the intention and essence of the project (Figures 6II and 7II).

Comprehension of the target considers the program and planning constraints, site constraints, and Serpentine Gallery's purpose [70]. In the case study, the relationship between the formal consideration adapted to the natural environment and the conditioning factors of the site (orography, in-between spaces, and accesses) is shown to be fundamental (Figure 6, site constraints). The relationship between the floor and the wall to enhance the park views will be decisive in the subsequent study of the supports (Figure 6a, relationships with site). The Pavilion is not conceived as an autonomous building; the concept is architecture itself [70]. The relationship between structure and envelope is defined as simple as it does not require a great deal of performance from the enclosure (Figure 6I). The relationship between form and structure is fundamental in the conception of the mesh, as well as its adaptation to the site and the relationship between form and geometry that emphasizes the sensations of dynamism, rhythm, and movement required. The relationship

between form and material is marked by using materials in harmony with the natural environment, with choices oriented toward natural materials such as wood (Figure 6II).

There is a need for further studies into reciprocal frame structures and traditional wood connections, as well as a review of the properties of this material (Figure 7II). It is also necessary to review historical constructive solutions [62] (Figure 7II(a–c)) that support the influence of *Arte Povera* [41] and serve as a counterpoint to previous editions of the Serpentine Gallery Pavilion, which were usually high-tech (Figure 7II). Regarding on-site assembly, the means and possibilities for offsite manufacturing must be reviewed (an objective that may be considered somewhat contradictory to the commitment to more traditional solutions).

All this is accompanied by an important objective, that of achieving a solution of great architectural interest. Such a solution will enhance the role of the Pavilion as an experimental architectural laboratory.

The idea, with this approach, serves as a guide, helps us to define the concepts and relationships that we must promote in the development of the project, marks the principles to be followed in the design process, and identifies a series of operations to be carried out. That is to say, the idea organizes the process, giving shape to a plan.

In the diagram, the different states of equilibrium of the form are reflected throughout the design process of the Serpentine Gallery 2005 (Figures 8I–III and 9IV). This allows us to take a reading of the variables considered and the response (adjustment) from the formal configuration. There are four stages shown in the diagram. The initial formal configuration is a rectangular structural grid with a flat roof that adapts to the characteristics of the environment and the existing trees (Figure 8a,b). Balmond [29] uses patterns to offer structure to the realm of ideas (Figure 8II(c)). In the next stage, Balmond's intervention and his interest in achieving dynamism in the structure [70] determine an adjustment to the form, resulting in a formal distortion of the roof. A roof appears that is no longer flat, but curved (Figure 8II(d)). The third stage adjusts and defines the perimeter curvature and corner elimination [70] (Figure 8III(e,f)). In the fourth stage, Balmond studies the formal configuration of the enclosure and the supports (Figure 9IV(d,e)). This is also a stage in which the first approaches to formal structural resolution appear. The process starts from a $3 \times 3$ grid pattern (Figure 9IV(a)), with an initial splitting operation of the grid elements transforming them into crosses (Figure 9IV(b)). Then, these crosses are displaced to achieve dynamism (Figure 9IV(c)). Possible joint solutions (Figure 9IV(e)) are still incipient, but the joints studied in the historical analysis of reciprocal frame structures can be recognized (Figures 3–5). The form will continue to evolve and adjust in successive stages involving the material, geometry, and function. These adjustments are reflected in the diagrams of the following stages (Figures 12–18).

The image is part of the sequence but is not subject to a discussion from the perspective of the analysis carried out. This is since the consideration of the proposed image from this approach is not the origin of the creative or design process, but rather with the image as the product of the creative work (Figure 10).

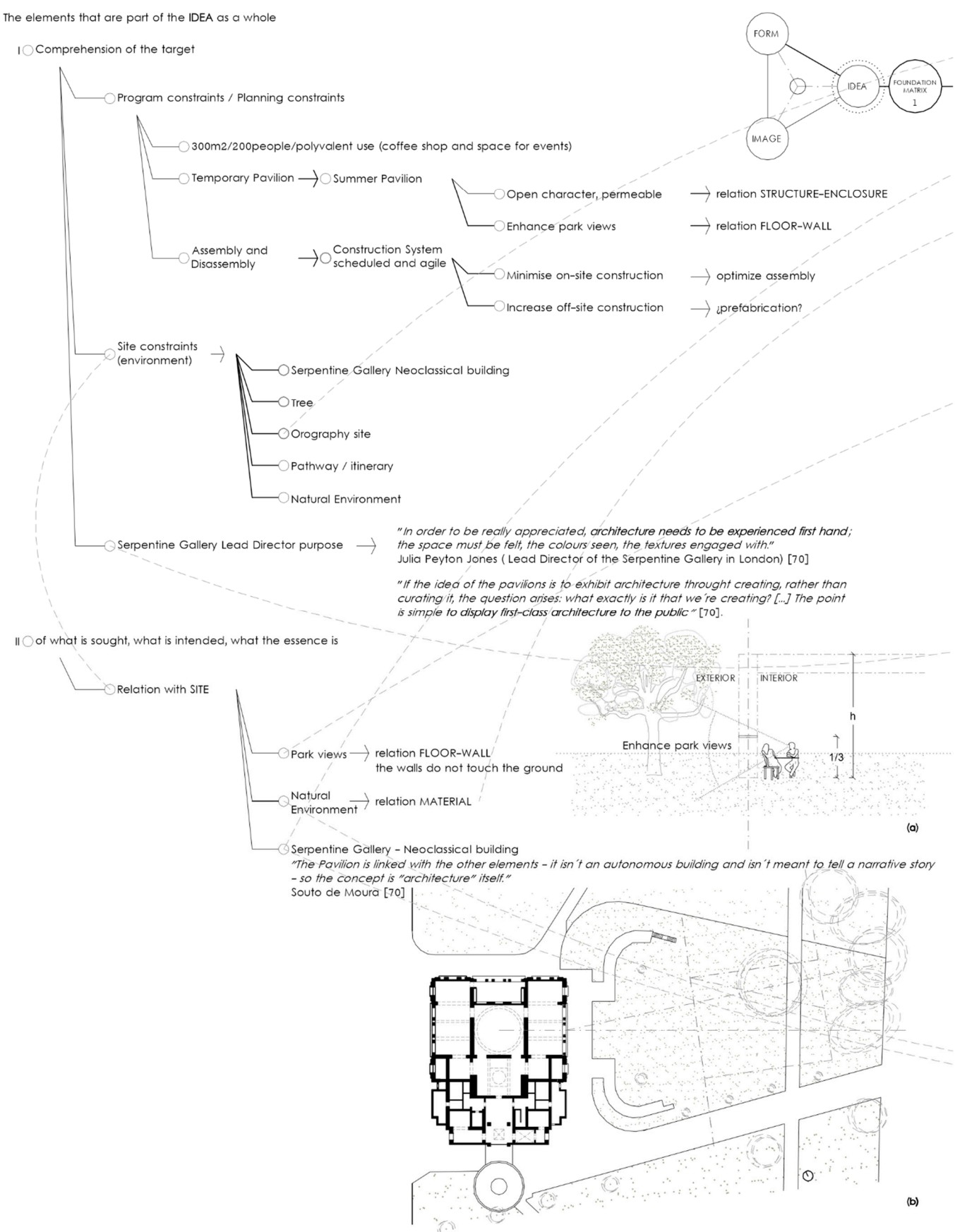

**Figure 6.** Foundation matrix 1.

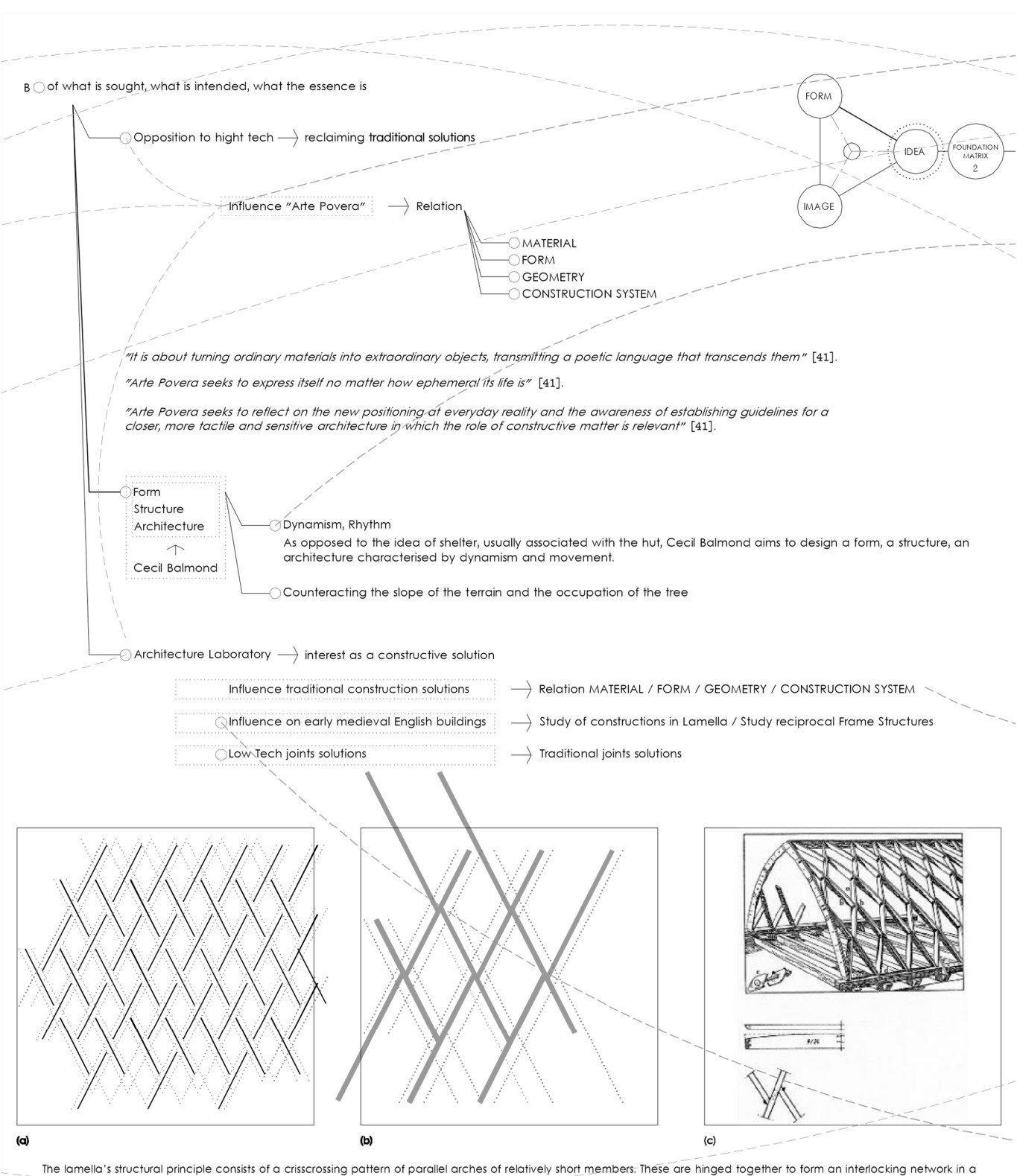

**Figure 7.** Foundation matrix 2. (**c**) Zollinger lamella roof drawing [71].

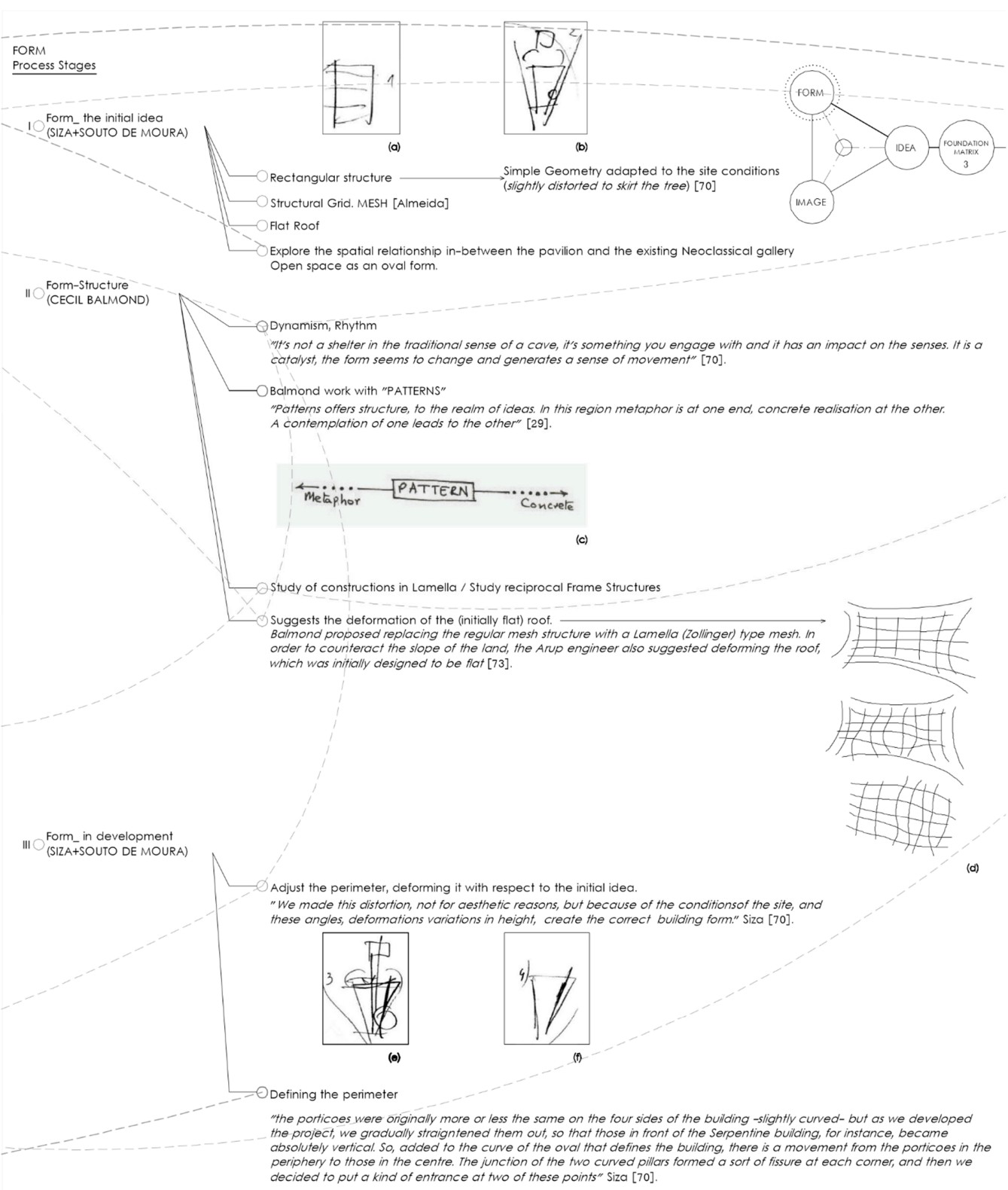

**Figure 8.** Foundation matrix 3. (**a**,**b**,**e**,**f**) Souto de Moura sketches cited in [72]; (**c**) Balmond scheme [29]; (**d**) del Río-Calleja based on Balmond sketches [72].

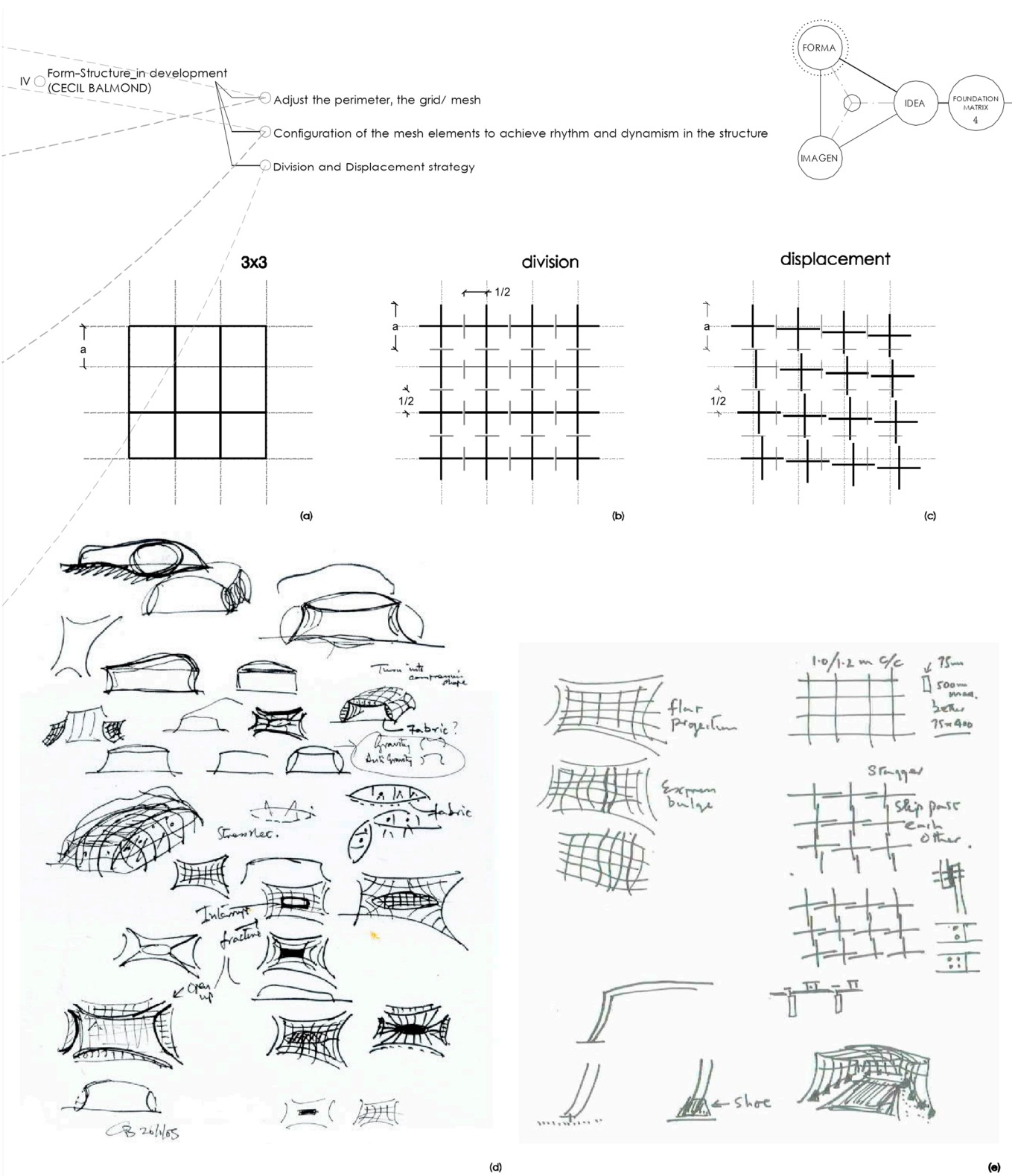

**Figure 9.** Foundation matrix 4. (**c**) del Río-Calleja interpretation based on Balmond sketches [27]; (**d**) sketches by Balmond [70]; (**e**) sketches by Balmond [27].

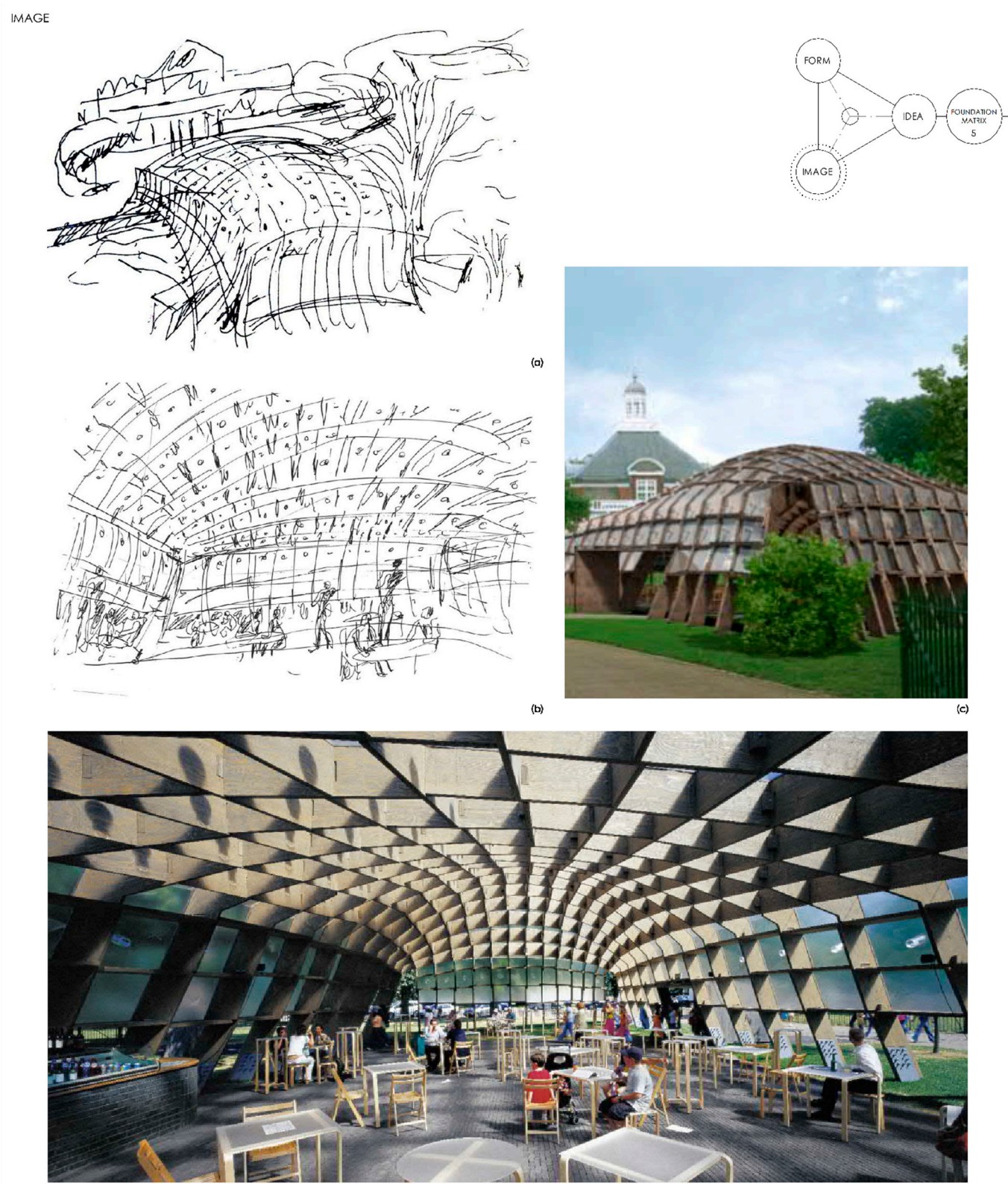

**Figure 10.** Foundation matrix 5. (**a**,**b**) Sketches by Siza [72]; (**c**,**d**) Serpentine 2005 images [73].

As a result of the foundation matrix, the target configuration, an organizational plan, is obtained. This is related to the constructive logic matrix at the level of comprehensive understanding of the concepts that form it, as shown in the workflow diagram in Figure 2. We can identify the inputs of the foundational matrix to the constructive logic matrix. From

the integral level of this matrix, these will reach the concepts of geometry, material, and function (Figure 11).

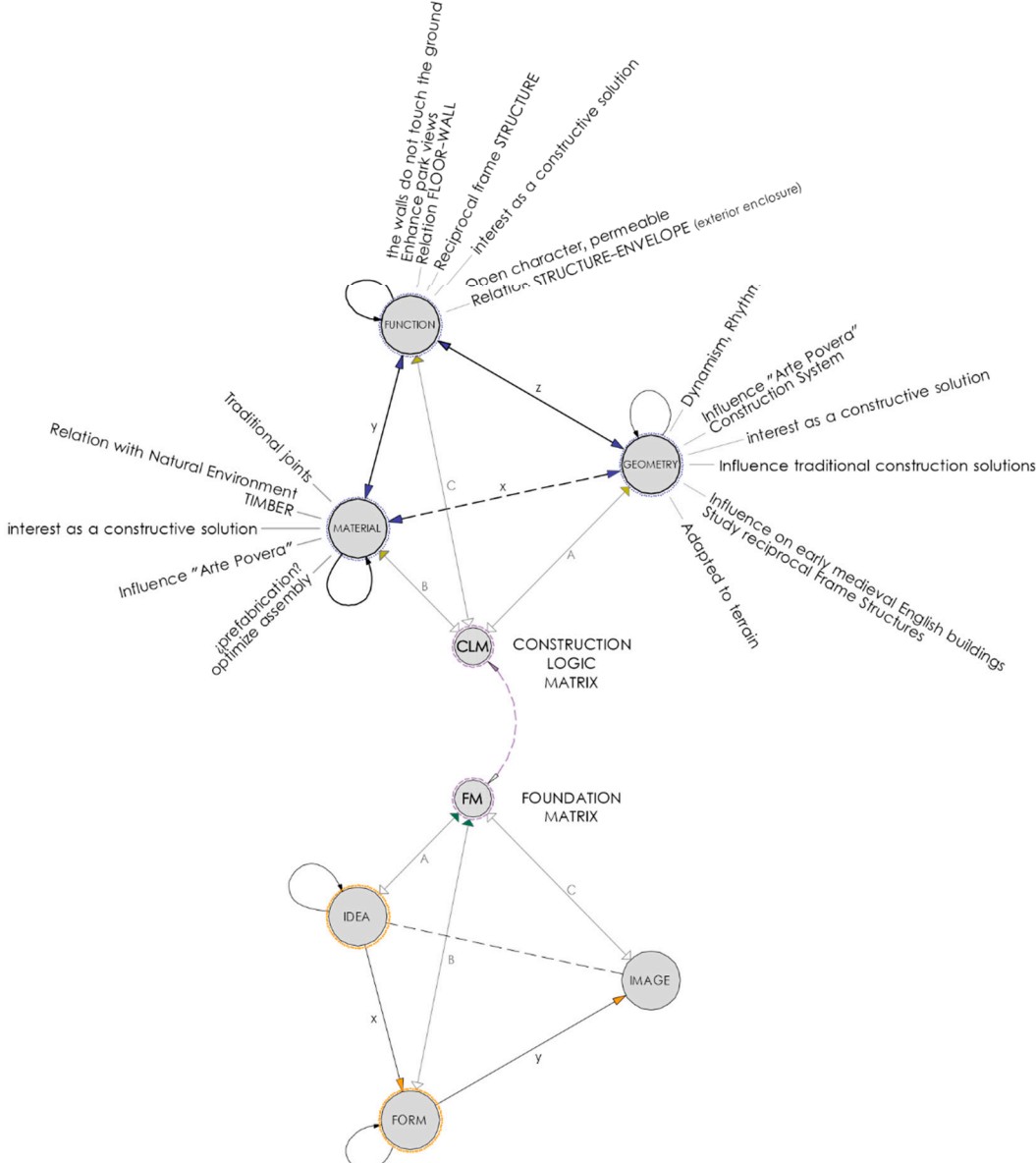

**Figure 11.** Matrices 3D model workflow and inputs from FM to CLM. Orange (x, y) for interaction between Idea, Form, and Image. Dark green for connections (A, B, C) from Idea, Form, and Image to FM global approach. Purple for relationships within matrix global approach. Light green for relationships (A, B, C) from CLM to its components. Blue for interactions (x, y, z) between Geometry, Material, and Function.

### 3.3. Systemic Analysis of the Constructive Logic Matrix

The matrix is made up of the concepts of function, geometry, and material and its relationships are complex because of its systemic behavior.

From a geometry perspective, the processes of analysis and diagramming initially address the issue of proportion. A first approximation of the geometrical definition of the Serpentine Gallery 2005 mesh has, as its starting point, a rectangle of Cordovan proportion, in which the short side corresponds to the side of the octagon inscribed in a circle, where the radius is the dimension of the long side (Figure 12I study of proportion).

The formal question should concern magnitude and direction terms. Starting from the basic four-element fan and the usual ratio of 1/3, the geometrical configuration of the mesh is studied (Figure 12II study of fan and mesh). The aim is to understand how it has been generated. To do so, it is necessary to find and determine the operations that govern its formal configuration. The diagram (Figures 12 and 13) shows the result of a process of searching for alternatives. Multiple combinations have been evaluated until a result has been reached that fits the case study grid. This analysis builds on the knowledge gained from a diagrammatic analysis of historical solutions (Figures 3–5) and a review of the relevant scientific literature on contemporary reciprocal structures [34,74–78].

The mesh configuration results from the combination of two operations: symmetry and displacement. Starting from the base module or fan, by performing symmetrical operations on the x- and y-axes, alternately, and selecting displaced axes, as the mesh enlarges or grows, an adjusted configuration is obtained (Figure 12II(A,B)).

The dynamism and movement that the Pavilion is intended to endow materialize in the configuration of the mesh. The aim is not only aesthetics; at the same time, these should also be part of the constructive solution. This is achieved by means of the displacement operation. This operation was introduced in the mesh developed by Zollinger in 1924 (Figure 4e–g). The displacement facilitates the resolution of the joint between the elements. The displacement proposed by Balmond (Figure 9a–c) generates a distortion of the mesh, which, when accumulated over its entire surface, loses dynamism (Figure 12II(C)). Adjustment is necessary. This consists of alternating the displacement to counteract it and avoid accumulated distortion (Figure 12II(C) adjusted displacement).

To transfer the displacement operation to the reciprocal mesh, an abstraction of the base module or fan is elaborated. This consists of grouping the four elements into two pairs (Figure 13A,B). This gives two displaced squares (Figure 13C). Each contains two elements that form a T. The sum of the two displaced squares is the base module or fan (Figure 13C). From this abstraction, the configuration is reproduced with the alternate displacement proposed above (by doing so for each of the squares). Two meshes are generated, one of square type 1 (Figure 13D) and one of type 2 (Figure 13E). The mesh growth law is the same in both cases. By overlapping the two meshes, the geometric configuration of the case study mesh is obtained (Figure 13F,G).

Once there is an understanding of and operations for how to generate the mesh configuration itself, the mesh is adapted to the limits and curvatures of the concrete form of the Serpentine Gallery 2005 (Figure 13III). The resulting mesh is distorted by two objectives identified in a systemic analysis of the foundation matrix: the adaptation to the intended shape of the double curvature and the release of the corner in the resolution of the supports.

In the case study of the Serpentine Gallery 2005, the choice of material was conditioned by the idea (Figure 7). It was also determined by the influence of *Arte Povera*, by the idea of the relationship of the Pavilion with nature, in a kind of mimesis, and by the influence of traditional farm buildings in England. The master plan of the idea studied reveals that the designers' intention was to build the Pavilion with timber. This choice greatly reduced the study of variables.

The properties of timber as a material [41] were reviewed, as well as general design considerations [79–87]. These would determine the constructive solution, in which the advantageous material properties were enhanced. Unsuitable properties for resolution also needed to be considered, with the potential to highlight a need for corrective measures to avoid design problems. Structural and geometrical constraints discouraged the use of sawn timber and led to an analysis of alternative timber products. A review of the properties of these products was carried out to guide the selection of the most suitable material for the aforementioned conditioning factors (Figure 14I), which were fundamentally those derived from the geometry, form, and structural system of the reciprocal frame structures.

The structural system chosen was characterized by high shear stresses located at the joints between the elements. The span to be bridged placed considerable demands on the material strength. The form required the construction of parts that reproduced the

curves. The planned assembly called on computer numerical control (CNC) machines for the cutting of the elements [88,89].

After an evaluation and study of the products, it was evident that the material with the most suitable behavior to meet the requirements was laminated veneer lumber (LVL), in particular, Kerto LVL grade Q. A panel of this is made of 3 mm-thick strength-graded softwood veneers, of which approximately 20% are oriented in a crosswise direction. This product had a higher shear strength than the rest of the materials evaluated. It is suitable for solving large spans due to its stiffness, strength properties, and light weight. In addition, it allows a slender proportion to be maintained in the elements that form the structure (Figure 14II).

From a function perspective (Figures 15 and 16), the case study was approached as a relationship of functions, referring firstly to the relationship between structure and configuration, and secondly, to the relationship between structure and enclosure (Figure 15I,II).

The relationship between structure and envelope is not reviewed in this case study as the analysis of the foundation matrix (Figure 6I) showed the low demands placed on the enclosure.

The case study, when referring to the relationship between structure and configuration, led us to analyze the structural behavior of the elements that made up the reciprocal structure [90]. From a functional perspective, the structure is understood as a system of supports characterized by how and what they support.

In a structural configuration, it is common to separate the functions of supporting and being supported. These functions are generally performed at discrete positions on an element. In the case of reciprocal frame structures, the functions are concentrated at the ends of the elements that make up the mesh. That is to say, all of the elements making up the reciprocal structure must simultaneously support and sustain another element (Figure 15a). As such, all the elements that make up the mesh have the same function, and there are no differential structural behaviors, except for the edge elements. This intrinsic characteristic of this type of structure contrasts with the usual structural configurations and makes it difficult to intuitively understand the structural behavior of reciprocal frame structures.

We were, therefore, conducting a systemic analysis of a structural system where its behavior was also systemic. As such, it was complex work to establish a hierarchy and functional organization of the elements that made up the structure.

Kohlhammer [90] developed an understanding of the behavior of reciprocal structures in a systemic way. They divided the system into hierarchical components: single bar, basic system, component system, and entire system (Figure 15A). System loading is seen as an iterative process of the interaction of the subsystems (Figure 15B), in which each iterative step is representative of the static equilibrium state of the subsystems. In each of these iterative steps, the supporting forces from the conditions of equilibrium result from the progressions of the sub-system observed. In the same way, the following subsystems continue to be analyzed. Kohlhammer [90] observed two types of progressions resulting from iteration: cyclic and diffuse (Figure 15C).

At the ends of the elements, the highest values of shear stress in a fan and bending moment in the supporting fan are concentrated. The value of the shear stress, as reported by Popovic [66], is related to the inclination of the mesh and the dimension between the supports that make up the base module or fan. Shear stress loading can be critical as timber, in general, has a relatively low shear strength.

The constructive definition at this point is so important that, as Pugnale [33] states, in reciprocal frame structures, the joints contribute directly to defining the structural behavior of the mesh.

To address the resolution of the joint, design considerations were set out (Figure 15). Traditional timber joints were studied from the relational perspective of functions (Figure 16), reaching a relational understanding of the joint resolution problem [91–97].

To reason around the conceptual resolution of the connection between elements developed and executed in the case study, the material and geometrical perspectives already discussed were also considered. It was necessary to change the perspective of analysis, to continue it in an overarching way. The overall analysis perspective is what we call constructive logic, which considers the three previous perspectives in an integral way (Figures 17 and 18).

The designers' intention was to resolve the joint by using a traditional type of joint. Structural requirements favored the execution of a traditional mortise and tenon joint to allow the transmission of shear stresses. In this type of joint, the connection between elements is made by machining, and the stresses are transmitted through compressions located in the perpendicular direction of the fiber, both in the mortise and tenon. These compressions are resisted by the crosswise-oriented (20%) veneers of the Kerto Q (Figure 14II).

In the joint, three elements converge in the same place. Its constructive resolution is the product of a strategy of combined operations.

The mortise is placed in the middle of the cross-section of the beam or element, which is where the value of the bending moment is at its minimum (with its maxima in the upper and lower parts) (Figure 17A).

At the center of each element are the ends of two other elements, with both perpendicular to the main one, and one on each side. Their axes are displaced, and this discontinuity simplifies the resolution of the joint while favoring the dynamic character of the structure. The displacement of the axes means that we can run a single larger mortise that receives the tenons of the two elements, one on each side (Figure 17G).

The slenderness of the elements means that the depth of the tenon is very shallow. The high shear stresses at the ends of the pieces or elements require a high-strength section of the tenon, meaning the depth of the tenons is not so important. The mortise and tenon connections transmit the stresses through the contact surfaces. Insufficient contact surface leads to stress-transmission problems. To increase the contact surface of the element, the resolution strategy is to consider the joint as a whole; if we can increase the contact surface of one of the elements, we will compensate for the reduced surface of the other.

In our work, this meant differentiating the ways in which they supported each other. The tenon of the first element was short and flush with the plane. The tenon of the second element was elongated, increasing its surface area. The surface of the larger tenon came into contact with the edge surface of the short tenon element, increasing the stress transmission capacity by friction. To hold the edges together, a metal pin-type joint, namely a bolt, was incorporated. This fitting, which ran through the timber pieces, was subjected to bending, generating crushing stresses in the timber pieces it joined, achieving solidarity between the elements that prevented them from separating (Figure 17a,b).

In this combined strategy, the joint was solved through the adaptation of various types of joints: mortise and tenon assembly, coupling, and pin-type metal joints. With this combined strategy, the solution held aesthetic value and also favored onsite assembly. In the geometrical definition of the solution, which did not take into account the formal distortions of the pavilion roof, the elements were equal and always had one short and one long tenon end, maintaining the characteristic structural behavior of reciprocal frame structures, in which all the elements have the same structural behavior (Figure 18a,b).

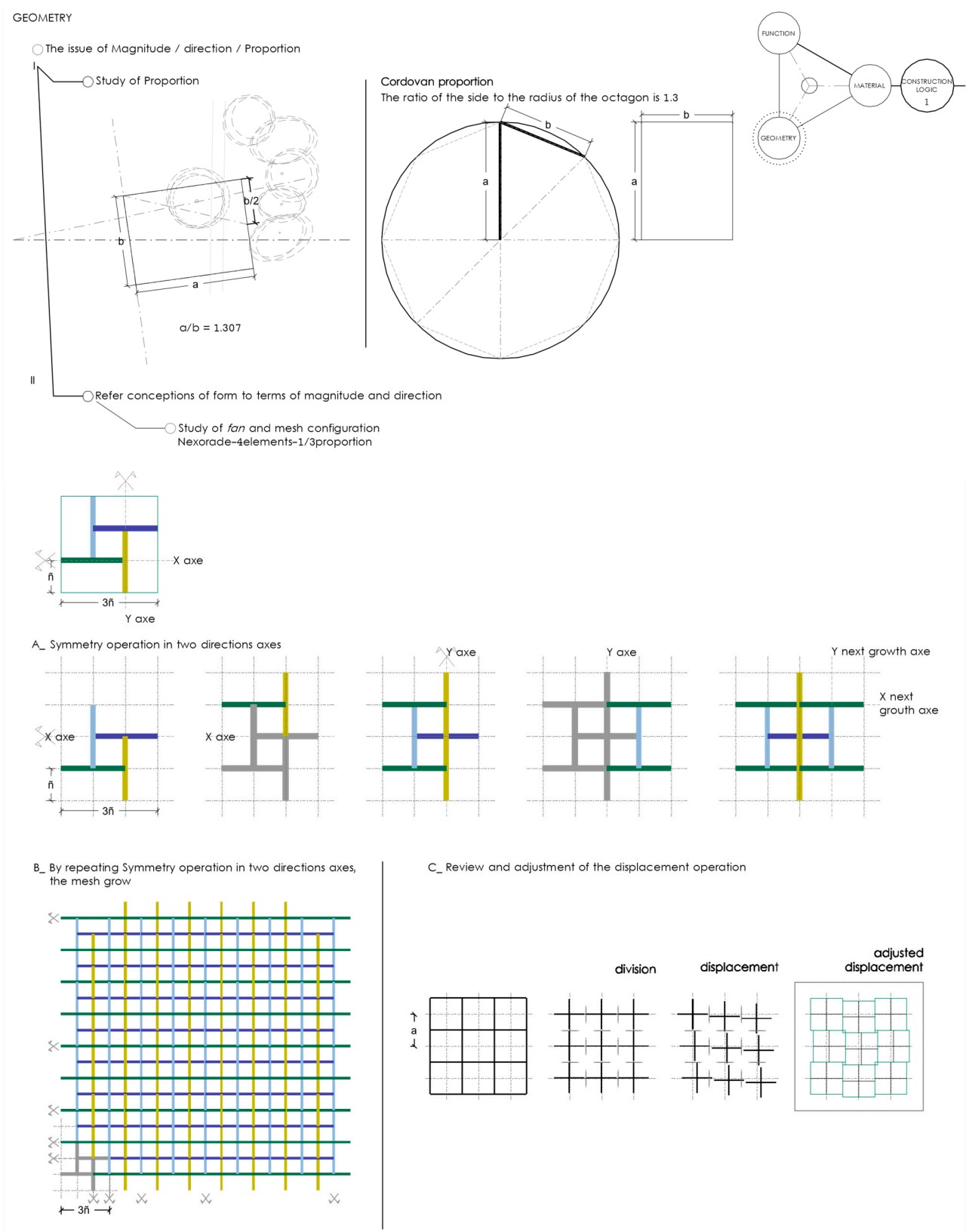

**Figure 12.** Construction logic matrix 1.

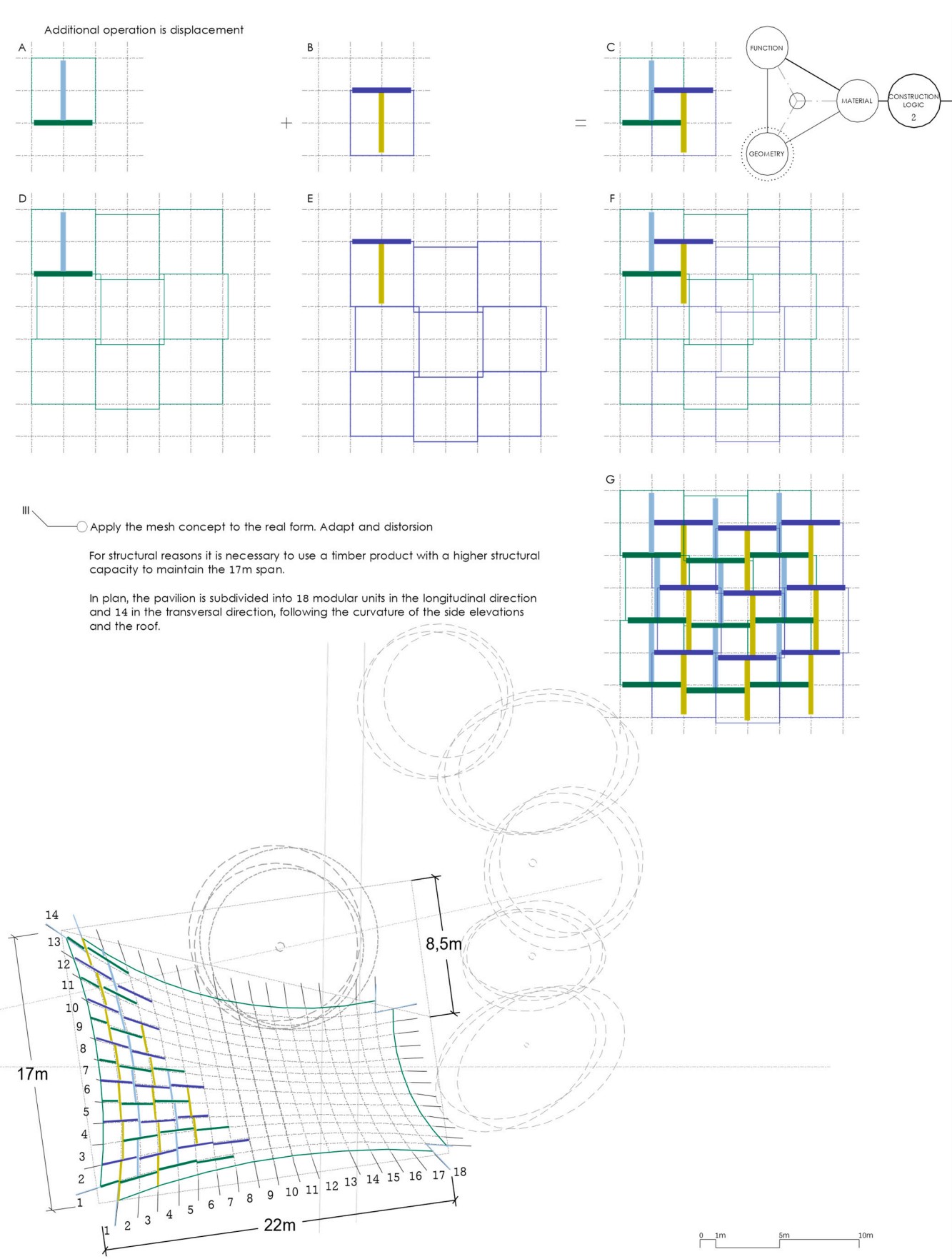

**Figure 13.** Construction logic matrix 2.

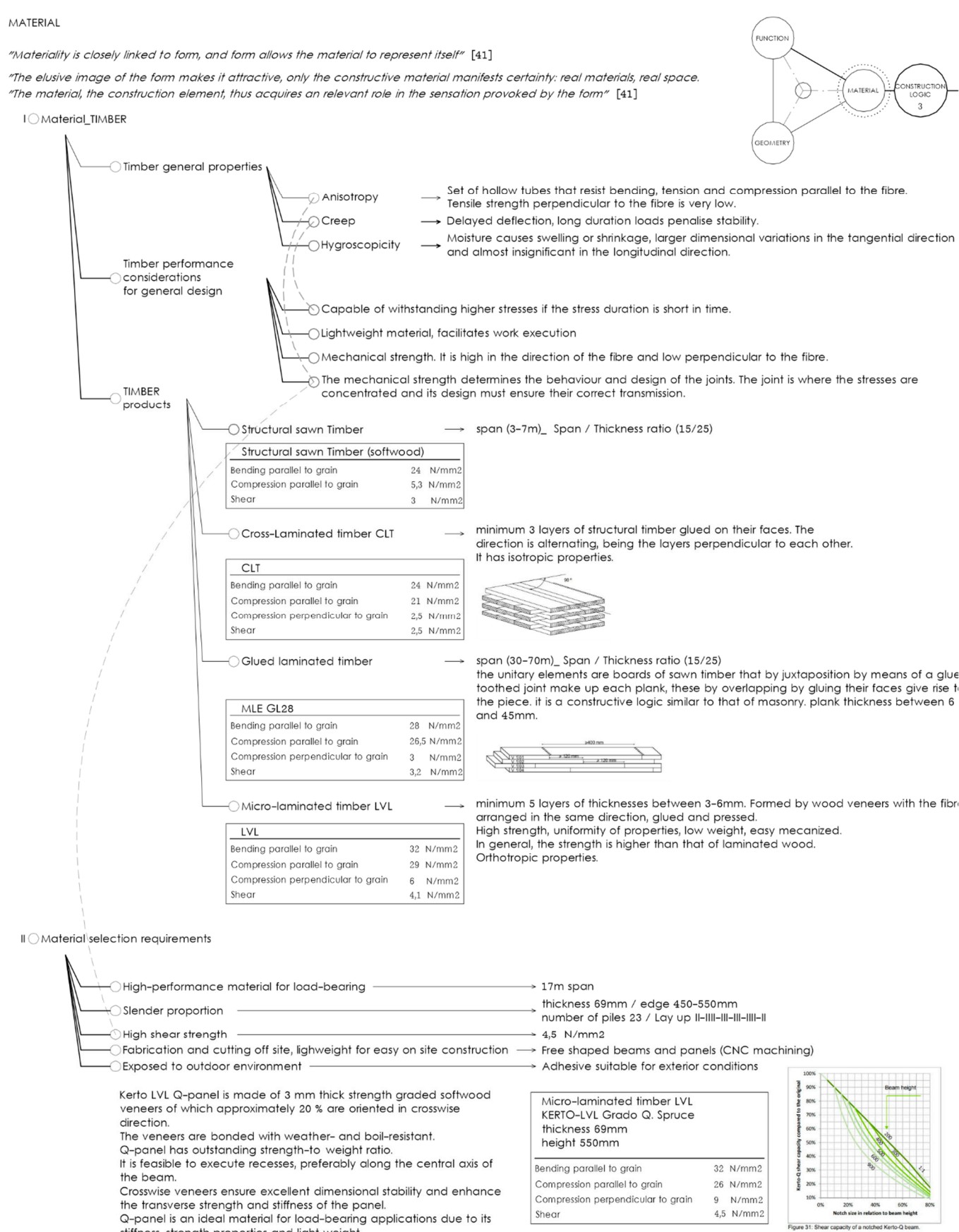

**Figure 14.** Construction logic matrix 3.

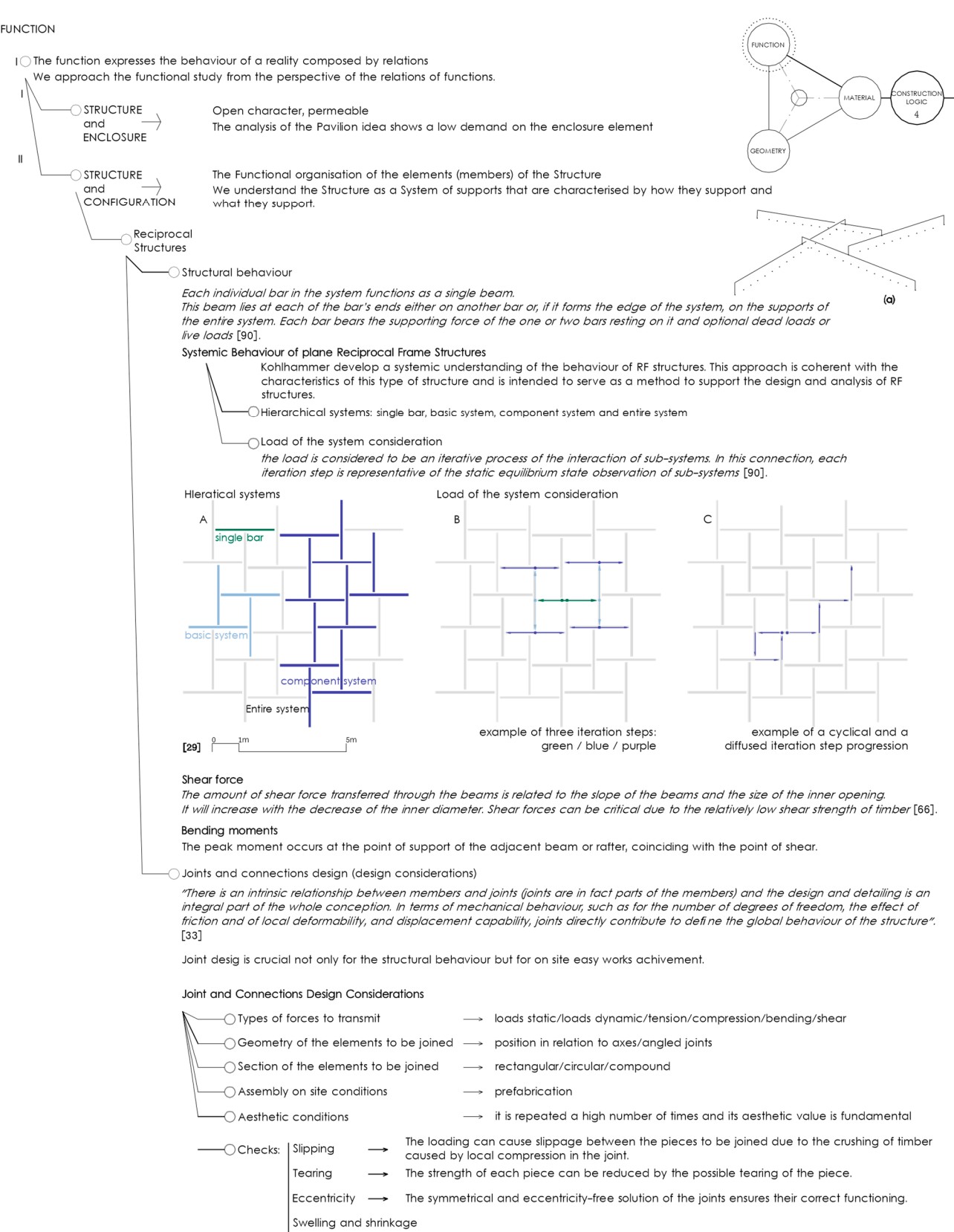

**Figure 15.** Construction logic matrix 4.

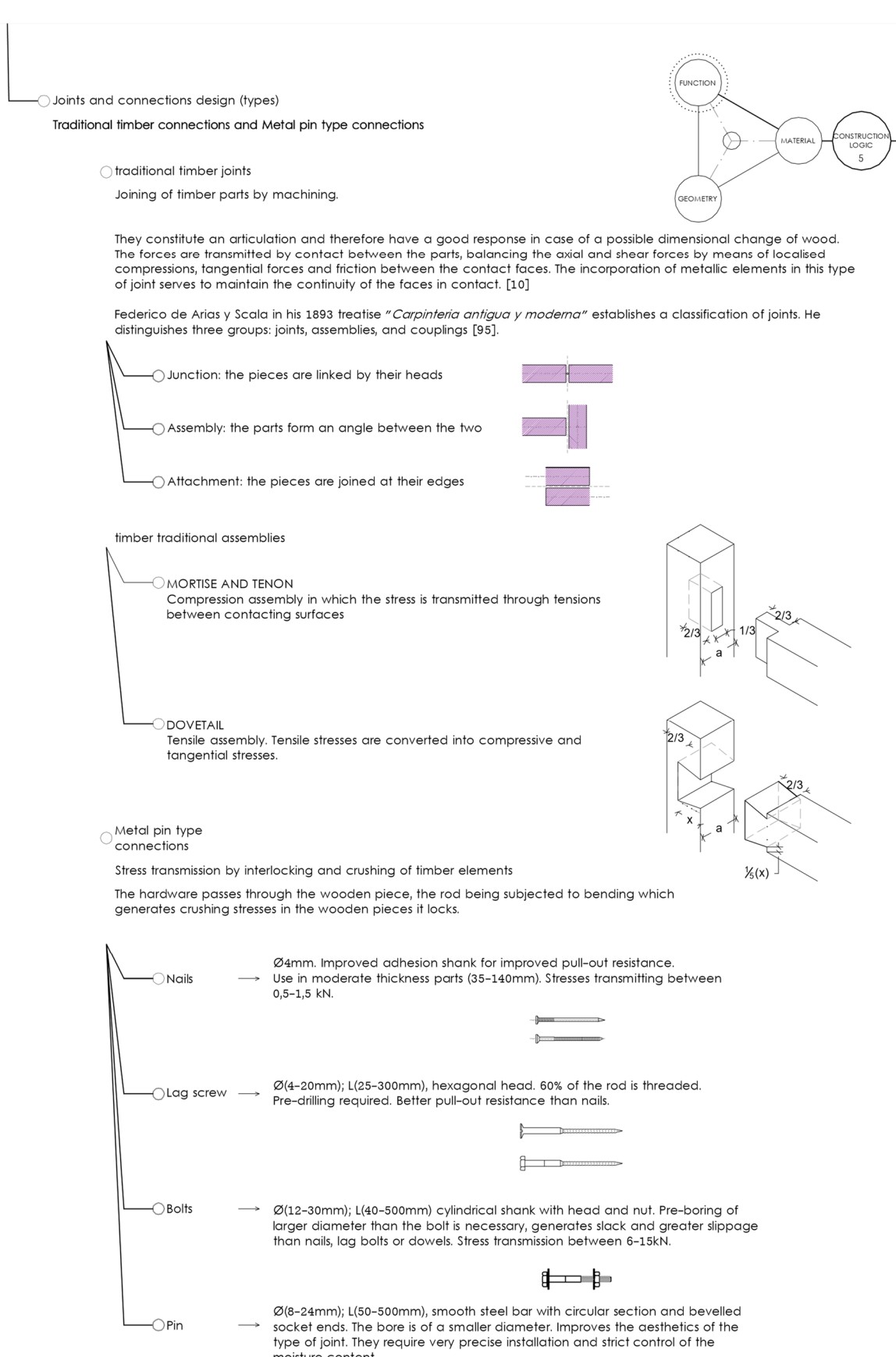

**Figure 16.** Construction logic matrix 5.

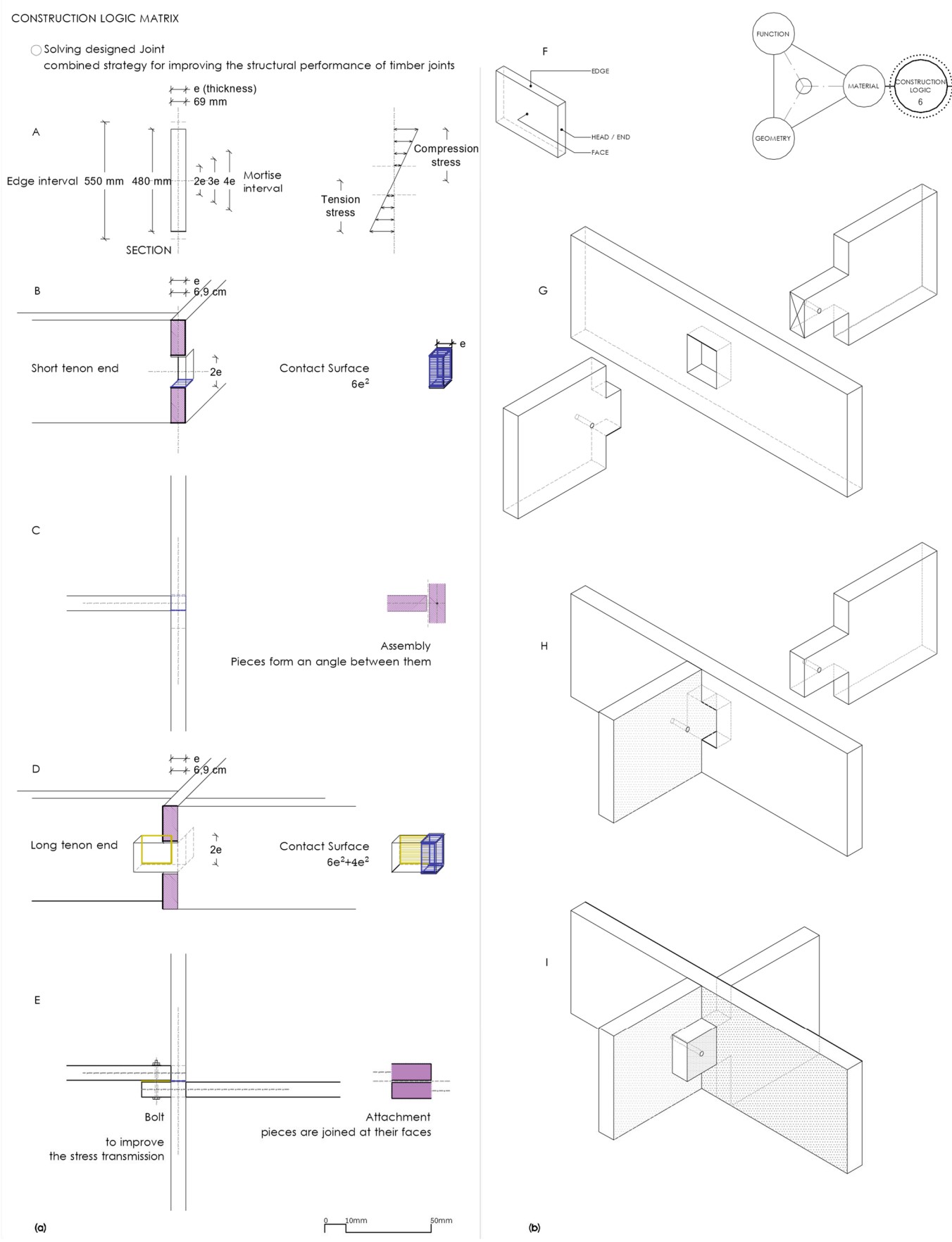

**Figure 17.** Construction logic matrix 6.

CONSTRUCTION LOGIC MATRIX

Structural behaviour of mortise and tenon joints:

Feio [96] studies the mortise and tenon joint, concluding that the parameters that have the greatest effect on the mortise and tenon joint are the compressive strength perpendicular to the grain and the normal displacement that can occur between the elements in contact. The compressive strength parallel to the grain does not influence the structural behaviour of this type of joint.

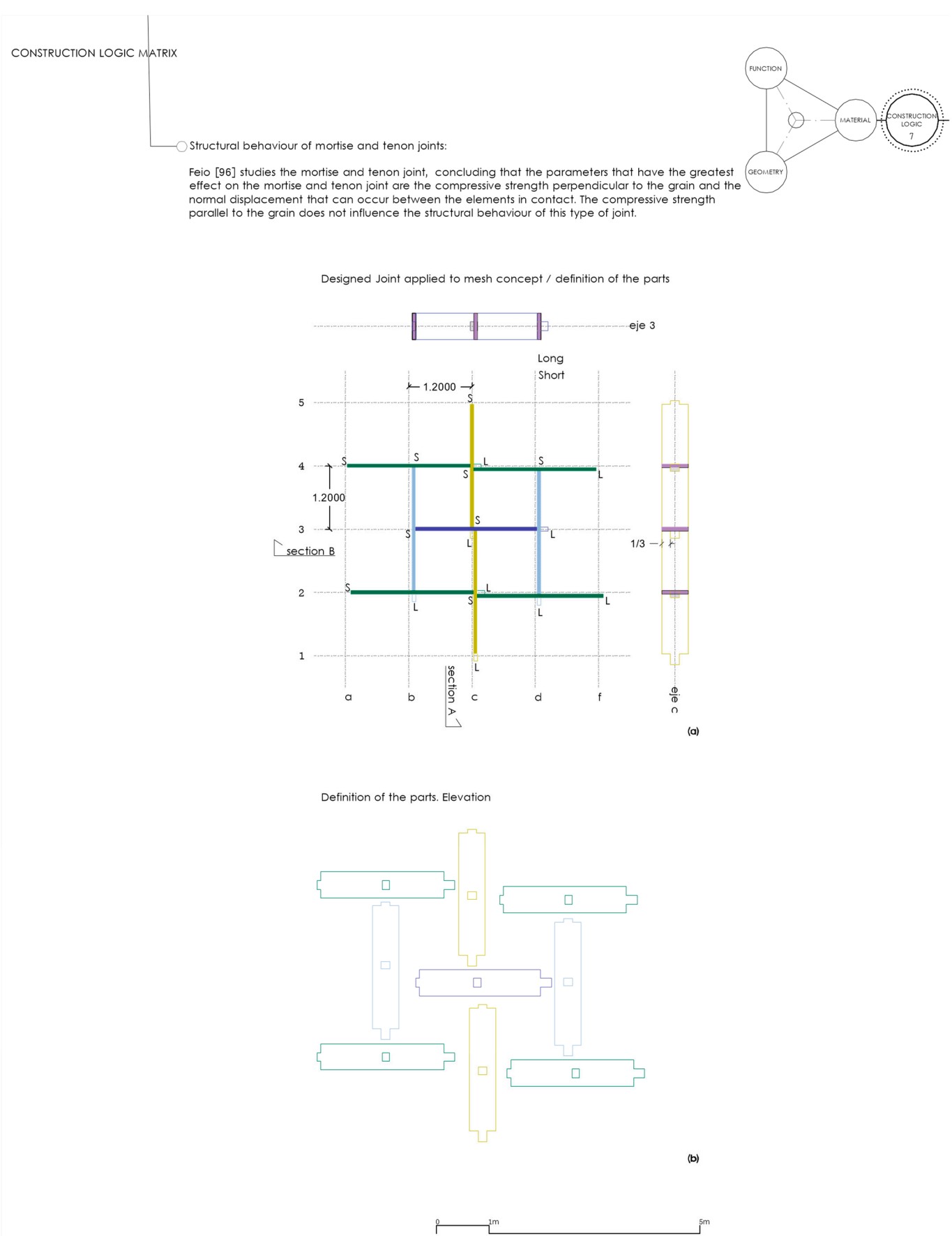

**Figure 18.** Construction logic matrix 7.

## 4. Conclusions

The relevance of the knowledge inherent in historical building solutions was demonstrated in this article. Such knowledge acquired by the authors, through a study and analysis of these solutions, enabled us to identify the presence of resolution strategies that have evidently been applied throughout the history of construction. The systemic analysis carried out allowed us to recognize in the reciprocal mesh configuration of the Serpentine 2005 an adaptation and combination of the Wallis and Zollinger solutions. The configuration of the mesh elements was influenced by one of Wallis' resolutions, while the joints were resolved based on the idea of displacement introduced by Zollinger in his Lamella-type structures. The history of construction is of paramount importance in architecture, and a thorough knowledge of historical solutions is essential. Once this is acquired, we can adapt it to other forms and/or transfer it to other materials or manufacturing technologies. The recognition of solutions from construction history in the *Summer Pavilion*, a symbol of contemporary architectural experimentation, underlines the relevance of the history of construction in architecture.

Our analysis of the case study was carried out using two matrices: foundation and constructive logic. These matrices form part of a model that attempts to establish a useful organization and hierarchy in the systemic approach to architecture. The model aims to identify and measure complex systemic relationships. This model is still under development by the authors, with the matrices developed and employed in this paper forming part of it. The systemic condition of the model requires iterative work on its development, with testing a concrete case.

Verifying the two matrices by testing them on the case study enabled us to recognize the systemic behavior of the approach. The definition of boundaries and concrete variables (those of the case study) that highlight the relationships between concepts facilitates the testing of the adaptive capacity of the approach and produces the intended relational knowledge. Testing the model on a case study garners more knowledge about the model itself, allowing us to complete its definition. Like systems, the model has an iterative process associated with its development.

The diagramming applied in the systemic analysis of the two matrices was a useful tool in the elaboration of structured representations of knowledge that facilitated the identification of concepts and relationships. It enabled us to maintain a holistic perspective on the approach (as a whole) at any time, which was fundamental for demonstrating the organization of knowledge.

The foundation matrix produces an understanding of the idea to be achieved, which serves as a guide in the decision-making process, helping to define the concepts and relationships that must be strengthened in the development of the project. This matrix serves to mark the principles to be followed in the design process and identifies a series of operations to be carried out. That is to say, the idea organizes the process by setting up a plan.

From the concepts belonging to the constructive logic matrix, it can be concluded that geometry serves as a support, a reference capable of establishing abstract and scale-free relationships to manage and coordinate an ideal model or archetype of reference. Thus, geometry constitutes an integral support for the building object. It is a support that grows and is enriched by exploring any aspect of the building's complexity, constituting a support for the architectural idea and the most powerful link between what we could call the architectural entity (which constitutes the essence or form of a building) and the constructive one. It also constitutes a methodological support, which, as a deductive method, offers logical support for the construction.

The material selected for the construction, Kerto LVL Q, is characterized by several qualities. When this material is systemically analyzed through the construction logic matrix, its qualities are transformed into material properties. From a function perspective, material is analyzed as a way of understanding the configuration and structural behavior of that which performs the supporting function. From a geometry perspective, material

is analyzed as a link between the architectural and constructive entities. The systemic approach, through the construction logic matrix, transforms material qualities into material properties. The idea of the structural element—as a resistant element—is conceived or generated when the material element is also part of a mechanical system (in this application, where the idea of a structural element is conceived while considering resistance). So, when we think of the material element as part of a mechanical system, we conceive of it as a structural element, and accordingly, assign it functions.

Systemic analysis from a functional perspective allowed us to understand the structural behavior of the reciprocal mesh that formed the roof in the case study. The systemic behavior characteristic of reciprocal meshes, with all elements understood to have identical structural functions, simplified the analysis from a functional point of view and highlighted the importance of joint resolution.

The resolution of the joint/junction problem was approached from a holistic view of geometry, material, and form. The requirements identified for the resolution came from the three concepts (perspectives) of the constructive logic matrix. After gaining an understanding of the requirements for the joint, we surmised that the joint combined solutions that were designed together. It is important to note that, in this case, none of the individual resolution strategies could solve the design problem. Only a combined view of the matrix and relational knowledge allowed the rationale for a joint solution that combined a mortise and tenon joint, attachment, and metal pin-type joint to be understood. The joint executed in this way in the case study did not renounce aesthetics. In its resolution, the displacement of elements was essential, which at the same time, brought a strong dynamism to the structure.

The systemic approach allowed us to understand the operations aimed at developing and defining the project from three perspectives (geometric, material, and functional) and at different scales (source-material, element, and building). From a system of relations, we determined the operations that generated the solution. Understanding the operations involved in that generative process, in an integrated way, leads to the acquisition of a type of knowledge (relational knowledge) that allows us to think about architecture in a way that affords us a greater capacity to conceive the building, approach it, design it, and solve it. Technical knowledge and artistic expression must be conceived as a unit. In a new perspective on how to manage complexity in architecture, a matrices check can be useful. Logic and creative thinking must remain integrated.

**Author Contributions:** Conceptualization, B.d.R.-C.; methodology, B.d.R.-C. and J.G.E.; diagrams, B.d.R.-C.; literature review, B.d.R.-C. and A.G.-S.; research and systemic analysis, B.d.R.-C.; writing—original draft preparation, review, and editing, B.d.R.-C. and A.G.-S.; supervision A.G.-S. All authors have read and agreed to the published version of the manuscript.

**Funding:** This research received no external funding.

**Institutional Review Board Statement:** Not applicable.

**Informed Consent Statement:** Not applicable.

**Data Availability Statement:** Not applicable.

**Conflicts of Interest:** The authors declare no conflict of interest.

## Abbreviations

The following abbreviations are used in the manuscript:

| | |
|---|---|
| FM | Foundation matrix |
| CLM | Constructive logic matrix |
| AGU | Advanced geometry unit |
| CNC | Computer numerical control |
| LVL | Laminated veneer lumber |

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
