# Peer review of "Architectural Systemic Approach: The Serpentine Gallery 2005, a Reciprocal Frame Case Study"

_buildings, doi:10.3390/buildings12071051_

Round 1

Reviewer 1 Report

Thank you for giving me the opportunity to read this interesting text. The work proposed is already substantial and the remarks made here are intended to provoke scientific emulation. The authors will have to answer some remarks, but others may be useful for other texts to come...

1. From the point of view of the content of the text :

- The interest of the text is to relate a theoretical systemic approach to the design process - a Foundation matrix (FM)/Construction logic matrix (CLM) model - with a concrete case study - the Serpentine Gallery Pavilion 2005. Although the systemic method is trendy, there are still too few case studies in the field of architecture to test the systemic concepts.
- Figures 12 to 18 are very interesting.
- In philosophy, the concepts of idea, form and image are the subject of many studies and theories, while the questions of the distinction between the intelligible, the materialized idea and the sensible reality is very difficult to establish. It is obviously not possible to give a state of the art here of all these concepts in philosophy, but perhaps a sentence should be added which indicates the voluntary reduction of these concepts to the specific framework of the study of structural systems in architecture? The same question arises for the concepts of intention and essence of the project, later in the article. The suggestion is not to define them in a general way, but to introduce them quickly in the context of the theme of the article.
- The concept of material appears twice, once on the CLM side (geometry, material, function) and once as an order (material, element, and building). This could create a conceptual ambiguity, should not another word be used for one of the two?
- In the theoretical model synthesised in figure 2, which relates the two matrices (FM/CLM): is the succession of idea, form and image really a sequence? Is it not rather three interrelated concepts, while pointing to the existence of a privileged pathway from idea to form, then from form to image? Even if the image is clearly considered by the authors as a final production, wouldn't the very existence of the image return on the form or on the idea? Similarly, in the proposed case study, a privileged path seems to be established from geometry to material, but the transfer to function seems less obvious? Moreover, additional relationships appear in the model of figure 11, notably with the image and function nodes. In the conclusion of the article, these achievements from the case study could perhaps allow to propose an update of figure 2 at the end of the article.
- In their next texts, the authors could perhaps give some additional keys to imagine the generalization of the model to all types of structures? Because the choice of reciprocal timber structures is a choice of linking systems thinking and structural design already initiated by research on tensegrity structures.

2. From the point of view of the form of the text :

- Some redundancies in the text.
- Some names need to be checked: it is not Sabastiano but Sebastiano Serlio, it is not Philibert L'Orme, but Philibert de l'Orme, De l'orme or Delorme.
- Between lines 109 and 134 of the article, the names of the works cited should be homogenised, perhaps separated by a semicolon, and the same form should always be used, e.g. Name of project/book by Name of author (year).
- For Figure 1 (Design Models Schemes), although they are indeed redesigned by Hugh Dubberly, I think the original authors of the schemes should be cited, such as Nigel Cross (2000) for the 3rd.
- In the manuscript uploaded to the platform, the numbers in front of the bibliographic references are missing. If they are not there, they should be added to enable readers to find their way around.

Author Response

Dear Reviewer 1:

Thank you very much for your kind comments. They are an encouragement and stimulus to continue working in this line of research. His knowledge of the subject is evident in his review work. It also shows that you have done a thorough reading of the article. We appreciate your time.

We agree with your observation on the insufficient presence of practical application on concrete cases of the theoretical approaches of systemic analysis in architecture.

We share the reviewer's view that it is difficult to synthesise important philosophical concepts in a text of this nature, addressed to the architectural scientific community. We consider that these concepts are succinctly referred to in the text and it is not our intention to enter into the realm of aesthetics in this article since, as it says, it would be outside the scope of our study. Following your recommendation, we add a sentence explaining this circumstance: “An in-depth philosophical discussion of the concepts of idea/form/image is not the subject of this article”. [lines 219 and 220]

Thank you for pointing out the redundancy in the concept of the material. To avoid possible ambiguity this term is adjusted. We employ source-material, incorporating a suffix when we refer to the scales considered in the use of the constructive logic matrix.

In the proposed foundation matrix, there is a strong linear sequence between the concepts of idea/form/image when they are treated independently. The three concepts interrelate with each other, but in the case of this matrix, they always do so from the global level. We have defined a higher hierarchical level in which this interrelation between the three concepts is considered. this global level to which we refer is represented graphically in the foundation matrix by the vertex of the tetrahedron in Figure 2 and Figure 11. We incorporate this paragraph into the text to clarify how these concepts interrelate at a more global level. [line 269-274]

Thank you very much for suggesting the extension of the case study to the field of tensegrity. We find this application of great interest and it will be a possible topic for future articles.

We have corrected the names of Sebastiano Serlio and Philibert de l´Orme, it was a transcription error.

The suggestion to homogenise the quotations between lines 109 and 134 is revised. For books or treaties is done according to the criteria: Author / Title / Date. [line 109-125]. For projects the criteria adopted is: Project / Author / Date. [line 128-135].   

Acknowledgement of the original authorship of the diagrams in Figure 1 is incorporated. [line 164]

Added the numbers of the references that were not included by mistake in the selection of the text style. [line 820-977]

Reviewer 2 Report

The article offers a detailed examination of a specific case study, using the concept of systemic approach in architecture and proposing an original set of analytical matrices (Foundation Matrix and Construction Logic Matrix). This approach leads to comprehensive conclusions that open up future avenues of enquiry, especially against the particular background of ephemeral architectural structures (Serpentine pavilions).

Still, one cannot help but wonder whether or not the analysis would have provided equally concrete results (especially in terms of the historic relevance of forms), had a less (or, for that matter more) 'suitable' case study been selected. Following up on this idea, Sou Fujimoto's project for the summer pavilion at the Serpentine gallery (2013) could offer an appropriate future case study for this research in progress, as a more integrated structure (in the sense that individual parts, such as 'walls' and 'roofs' are even less distinguishable than in Siza/deMoura/Balmond's project. In this same line of thinking, it could be worth better explaining the rationale that led the authors to identify the roof element as 'the most complex element' (line 295) of the project, for their testing.

Another weakness that could be argued with respect to the analytical framework, and more specifically to the proposed Foundation matrix is the rather casual dismissal of its 'image' component (lines 264-65). The authors explain that this is conventionally understood as related to the end of the creative process and for that, not included; nevertheless, to consider 'images' under an unconventional light (as part of the creative process rather than its end, since after all, they are consistently used also in the paper's diagrammatic analysis, as the authors themselves correctly point out, see lines 490-93) could provide grounds for more significant (and perhaps unexpected) conclusions.

The paper offers a comprehensive overview of historical precedents and is underpinned by well-selected references ranging from a broad range of fields. The study is presented in a well-structured manner; the definition of certain key terms, such as 'reciprocal' structures, could have been introduced earlier in the paper (for example, in the first part of the Introduction section) to facilitate the average reader.

The language is straightforward and to the point but a reviewing of the paper's paragraph structure, especially in its first two parts is recommended. At present, there are a lot of paragraphs consisting of a single, long phrase, which, in some instances makes the discourse read as fragmentary and difficult to follow.

Author Response

Dear Reviewer 2:

Thank you very much for the time you have taken to study our work in depth, as well as for the kind comments that help us to improve the academic quality of this work.

With a view to broadening the practical application of these systemic analysis matrices, the work suggested by the reviewer seems to us to be very appropriate. We will consider it as a possible case study for future work.

The consideration of the roof as the most complex element of the architectural work is supported by its repercussions both in treatises and in the discipline of construction history. In international congresses on the history of construction, roofs occupy a large part of the sessions. We would also highlight the recent publication of Gargiani: L'architrave le plancher la plate-forme (2012). In this book, under the direction of Gargiani, a hundred internationally renowned specialists describe the evolution of the constructive knowledge required for the execution of horizontal structures.

Regarding the issues raised by reviewer 2 on the workflow of the foundation matrix, specifically with the concept of image, we clarify that in the proposed foundation matrix, there is a strong linear sequence between the concepts of idea/form/image when they are treated independently. The three concepts interrelate with each other, but in the case of this matrix, they always do so from the global level. We have defined a higher hierarchical level in which this interrelation between the three concepts is considered. this global level to which we refer is represented graphically in the foundation matrix by the vertex of the tetrahedron in Figure 2 and Figure 11. We incorporate this paragraph into the text to clarify how these concepts interrelate at a more global level. [line 269-274]

The definition of Reciprocal Frame Structures is given in the introduction, lines [97-101]. For more information on the concept, we refer readers to the quotations in lines [101 and 103].
